# Causal Modeling of Selection in Evolution

**Haoyue Dai** [1]  **Zeyu Tang** [2]  **Peter Spirtes** [1]  **Kun Zhang** [1][3]

## Abstract

Understanding potential selection in data is crucial for causal discovery; we argue that "selection" in common narratives takes two forms, which we term *static* and *evolutionary* selection, respectively. Static selection refers to a one-shot filtering process where observed data consist of a *subset* of the population of interest, as in survey volunteer bias. Evolutionary selection, in contrast, operates through repeated rounds of differential fitness in reproduction, where observed data constitute the latest *generation* shaped by a historical trajectory, as in immune adaptation, antibiotic resistance, and social norm emergence. Existing methods largely conflate these two forms and rely on an identical graphical model of selection. We show that this model is valid for static settings but fails to characterize data under evolution, yielding false discovery results. To address this, we introduce a new model that specifically characterizes evolutionary selection, and develop a sound and complete procedure for identifying such models from data across one or multiple environments or generations. Experimental results validate the method's ability to uncover the relevant mechanisms underlying evolution from data.

## 1 Introduction

At the core of scientific inquiry lies causal discovery, the task of identifying causal relations from data (Spirtes et al., 2000; Pearl, 2009). This task is challenging, in part due to the fact that observed dependencies may not arise from causation alone. One key source of such dependencies, which is the focus of this paper, is selection, where data are preferentially observed through some systematic, but often unobserved, mechanisms.

[1]Carnegie Mellon University [2]Stanford University [3]Mohamed bin Zayed University of Artificial Intelligence. Correspondence to: Kun Zhang <kunz1@cmu.edu>.

*Proceedings of the 43rd International Conference on Machine Learning*, Seoul, South Korea. PMLR 306, 2026. Copyright 2026 by the author(s).

Selection plays essential roles in causal analysis. In some scenarios, it introduces a bias that must be accounted for. For instance, in political surveys, recruitment methods, e.g., by mail or by phone, may systematically favor individuals with certain education or income levels, leading to incorrect conclusions if selection were ignored (Groves, 2006). In other scenarios, instead of a bias to be corrected, the selection is itself an integral component of the data-generating process that needs to be discovered. For instance, in microbial experiments, observed dependencies between bacterial traits may arise from preferential proliferation of certain trait combinations rather than direct causal relations. Understanding such selection mechanisms is therefore crucial for downstream tasks, such as designing treatments (Dempster et al., 2021).

These considerations have motivated a substantial literature on causal modeling of selection. For causal discovery, one seminal contribution is the Fast Causal Inference (FCI) algorithm (Spirtes et al., 1995), which exploits conditional independence (CI) constraints in the presence of latent confounding and selection bias. Subsequent work has extended this framework to local (Versteeg et al., 2022), sequential (Zheng et al., 2024), and interventional settings (Dai et al., 2025a), as well as to additional parametric assumptions (Zhang et al., 2016; Kaltenpoth & Vreeken, 2023). For causal inference, the recovery of causal effects from selection bias has also been studied extensively (Bareinboim & Pearl, 2012; Bareinboim et al., 2014; Correa et al., 2019), with a close connection to the data missingness problem (Mohan et al., 2013). A detailed review is provided in Appendix C.

Despite their diversity in objectives and techniques, these works share a common graph modeling paradigm, illustrated in Figure 1a. In this paradigm, additional variables representing selection are directly added to the original causal graph over the observed variables. These selection variables are modeled as effects of the factors governing selection mechanisms, and are typically binary indicators of whether a data point satisfies the corresponding selection criteria. Observed data are then interpreted as the remaining variables conditioned on specific values of the selection variables (i.e., being selected). In this way, constraints induced by the graph under conditioning can be used (Spirtes et al., 1995).

However, does this simple and widely used paradigm always correctly capture data under selection? To answer this ques-

tion, let us reconsider the nature of selection itself. As we argue next, selection in common narratives actually takes two distinct forms. Distinguishing between them is crucial for examining when this paradigm applies, and when it does not.

The first form, which we term *static selection*, corresponds to selection occurring in a single shot, where a *subpopulation* is selected from the global population of interest. Static selection typically arises during data collection, as in the political survey example discussed above, and is often known as non-compliance and volunteer bias (Hernán et al., 2004). In such cases, the standard graphical models (e.g., Figure 1a) indeed provide appropriate representations. For clarity, we refer to these models hereafter as *static selection models*.

The second form, however, is more subtle. Turning to the other microbial example discussed above, the observed dependencies of certain traits do not arise from any one-shot selection at data collection. Instead, bacteria carrying advantageous traits exhibit higher proliferative fitness, producing more offspring that inherit these traits and are repeatedly favored over time. After multiple generations to provide sufficient material for sequencing, these fittest bacteria come to dominate the observed population. In this process, selection manifests through multiple consecutive rounds and is coupled with reproduction, rather than through a single event.

We refer to this second form as *evolutionary selection*. Unlike static selection, the observed data do not correspond to a subpopulation of the global population at any fixed timestamp. Instead, they represent a *generation* of individuals whose composition has been shaped by a historical trajectory of selections and reproductions. Each generation both reflects the confluence of past selection and serves as the substrate for future selection. Canonical examples include natural selection in biological evolution, such as immune adaptation, antibiotic resistance, and the well-known increase of dark-colored moths in Manchester due to air pollution during the Industrial Revolution (Kettlewell, 1955). More broadly, evolutionary selection also arises beyond biology whenever generational processes are present, for example in how social norms gradually and spontaneously emerge, and in how urban neighborhoods undergo gentrification with demographic shifts (Boyd & Richerson, 1988).

Then, can the existing paradigm of static selection models still correctly capture data under evolutionary selection? At first glance, the answer may seem affirmative: Consider again the microbial example. One might view bacterial traits by $X_1, X_2, X_3$ and fitness by $S$ in Figure 1a, and find that this static model is consistent with the dependence $X_2 \not\perp\!\!\!\perp X_3$ observed in data. It may thus appear that the statistical information (or at least the CI constraints) in data, even after evolutionary selection, can still be equivalently represented by a static selection model. Indeed, such static models have often been taken for granted in scenarios where selection is

in fact evolutionary (Versteeg et al., 2022; Luo et al., 2025).

Interestingly, however, this intuition is incorrect. As we will show in §2, static selection models fail to capture data under evolutionary selection and can thus lead to false discovery results. This is because reproduction, when coupled with selection, induces additional dependencies that cannot be characterized by a one-shot selection event, calling for another causal model that explicitly represents such processes.

Yet, to the best of our knowledge, no such model currently exists. While evolutionary biology has developed a rich collection of mathematical models such as Fisher's principle (Fisher, 1930), fitness landscape (Wright, 1932), and modern synthesis (Mayr & Provine, 1998), they are not formulated within causal frameworks. A more related literature is called evolutionary or reciprocal causation (Oyama et al., 2003; Uller & Lala, 2019), but it focuses primarily on philosophical aspects rather than formal graphical treatments. As a result, despite the ubiquity of evolutionary selection across disciplines, there remains no principled model or method for discovering its relevant mechanisms from data.

Bridging this gap is the goal of this work. Towards this goal, the remainder of this paper is organized as follows. In §2, we formally define the *evolutionary selection model*, a causal graphical model to represent data-generating processes driven by evolutionary selection. We use it to illustrate why existing static selection models do not suffice in evolutionary settings. In §3, we characterize the model's Markov properties, and develop a sound and complete identification procedure using CI constraints in data under evolutionary selection, with particular attention to result interpretation. In §4, we generalize to heterogeneous data such as observations from multiple environments or generations. We show conceptually how different selection mechanisms shape different communities via evolution, and algorithmically how such heterogeneity improves model identifiability. In §5, we validate the proposed method through simulations and real-world data across biological and social studies, showing its ability to uncover relevant mechanisms underlying evolution. We conclude in §6 with a discussion of limitations.

## 2 Evolutionary Selection Model

In this section, we introduce the new evolutionary selection model, a causal graphical model to represent data-generating processes underlying evolutionary selection, and then use it to illustrate why existing static models fail in such settings.

**Notations on graphs.** In a directed acyclic graph (DAG) $\mathcal{G}$, let $V(\mathcal{G})$ be its vertex set. For any vertices $a, b$, we say $a$ is a *parent* of $b$ and $b$ is a *child* of $a$, denoted by $a \in \mathrm{pa}_{\mathcal{G}}(b)$ and $b \in \mathrm{ch}_{\mathcal{G}}(a)$, when $a \to b$ is an edge in $\mathcal{G}$, written $a \to b \in \mathcal{G}$; $a$ is an *ancestor* of $b$ and $b$ is a *descendant* of $a$ if $a = b$ or there is a directed path $a \to \cdots \to b$ in $\mathcal{G}$, denoted by $a \in$

(a)  (b)

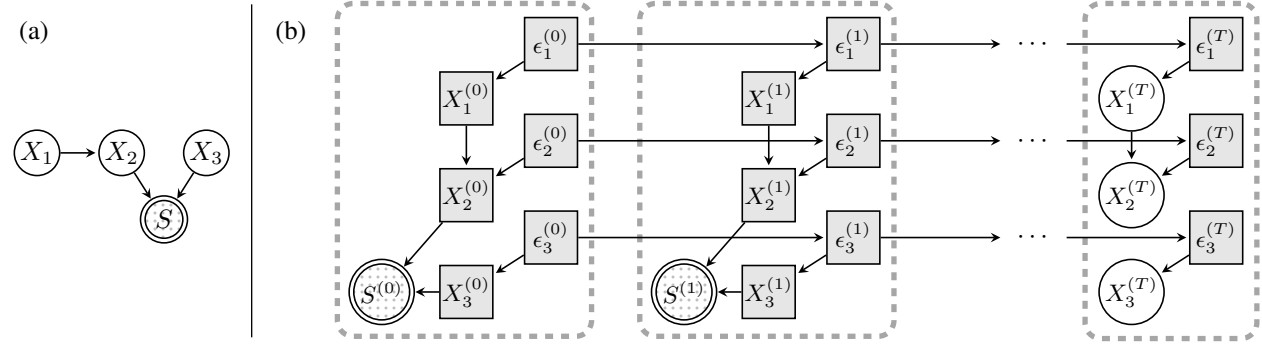

*Figure 1.* An illustrative comparison of the two selection models. Throughout, white circles indicate observed variables, dotted double circles indicate selection variables, and gray squares indicate unobserved variables. **(a)** An example causal graph, denoted $\mathcal{G}$, of the commonly used *static selection models*. **(b)** The corresponding *evolutionary selection model*, denoted $\mathcal{G}^{(T)}$, as defined in Definition 1, which represents the data-generating process of the observed variables $X^{(T)}$ under evolutionary selection. Dashed boxes indicate *generations*.

$\mathrm{an}_{\mathcal{G}}(b)$ and $b \in \mathrm{de}_{\mathcal{G}}(a)$. These notations extend to sets: e.g., for any vertex set $A$, $\mathrm{an}_{\mathcal{G}}(A) := \bigcup_{a \in A} \mathrm{an}_{\mathcal{G}}(a)$. We say an ordering $\pi$ of $V(\mathcal{G})$ is *topological* to $\mathcal{G}$, when $\pi(a) < \pi(b)$ implies $b \notin \mathrm{an}_{\mathcal{G}}(a)$. When we denote a set $A \subseteq V(\mathcal{G})$ as *selection variables*, for any $b \notin A$, we say $b$ is *involved in selection* when $b \in \mathrm{an}_{\mathcal{G}}(A)$; specifically, $b$ is *directly selected* when $b \in \mathrm{pa}_{\mathcal{G}}(A)$, and otherwise *indirectly selected*.

**Notations on evolution.** Let $X = (X_1, \ldots, X_d)$ be the observable variables of interest, such as phenotypic traits of a species. Observed data typically correspond to a *generation* of this species at a certain evolutionary time point; we explicitly denote by $X^{(t)} = (X_1^{(t)}, \ldots, X_d^{(t)})$ the traits of the individuals in the $t$-th generation (i.e., after $t$ rounds of reproduction). To avoid confusion with "parent" and "child" in graphs, we use the terms *progenitor* and *offspring* to describe the lineage relations between generations.

To model evolution, it is essential to understand how generations are causally related. When we say "taller individuals tend to have taller offspring," this does not imply a direct causal effect from a progenitor's height $X_i^{(t)}$ to the offspring's height $X_i^{(t+1)}$. Instead, the progenitor's height is affected by the related genes, which are inherited by the offspring and, in turn, affect their height. Such heritable factors, like genes in biological settings and family resources in social settings, are typically unobserved but play a key role in evolution. We thus denote these factors governing the corresponding variables $X^{(t)}$ by $\epsilon^{(t)} = (\epsilon_1^{(t)}, \ldots, \epsilon_d^{(t)})$, which, from a structural causal model view, serve as the exogenous noise terms for the $t$-th generation. Also, note that when $\epsilon_i^{(t)}$ directly affects $\epsilon_i^{(t+1)}$ during inheritance, their exact values may not be the same due to processes such as mutation.

With the above notations and motivation in place, we now formally define the evolutionary selection model, as follows:

**Definition 1** (**Evolutionary selection model**). Let $\mathcal{G}$ be a DAG with vertices $X \cup \{S\}$. The *evolutionary selection model* for a generation $T \geq 1$ is a DAG $\mathcal{G}^{(T)}$ with vertices:

- $X^{(0)} \cup \cdots \cup X^{(T)}$, the trait variables;
- $\epsilon^{(0)} \cup \cdots \cup \epsilon^{(T)}$, the exogenous heritable factors;
- $\{S^{(0)}, \ldots, S^{(T-1)}\}$, the binary reproduction indicators, where $S^{(t)} = 1$ indicates that an individual in the $t$-th generation has offspring, and $0$ otherwise.

$\mathcal{G}^{(T)}$ consists of the following four types of edges:

- $\{X_i^{(t)} \to X_j^{(t)} : \forall X_i \to X_j \in \mathcal{G}, \text{ and } t = 0, \ldots, T\}$, the direct causal effects among traits within generations;
- $\{X_i^{(t)} \to S^{(t)} : \forall X_i \to S \in \mathcal{G}, \text{ and } t = 0, \ldots, T-1\}$, effects of traits on previous generations' reproduction;
- $\{\epsilon_i^{(t)} \to X_i^{(t)} : \forall i = 1, \ldots, d, \text{ and } t = 0, \ldots, T\}$, the governing mechanisms of exogenous factors on traits;
- $\{\epsilon_i^{(t)} \to \epsilon_i^{(t+1)} : \forall i = 1, \ldots, d, \text{ and } t = 0, \ldots, T-1\}$, the inheritance and mutation of exogenous factors.

Accordingly, the observed data of the $T$-th generation is described by the distribution $p(X^{(T)}|S^{(0)} = \cdots = S^{(T-1)} = 1)$, which we abbreviate as $p(X^{(T)}|S^{(<T)} = 1)$.

An illustrative example of such evolutionary selection models (Definition 1) is shown in Figure 1b. In what follows, we explain various aspects considered in this model design.

**What is $\mathcal{G}$ used in definition?** The DAG $\mathcal{G}$ corresponds to a static selection model that one might consider as a first thought, where an additional variable $S$ intended for fitness is directly added to the original causal graph among $X$. Although, as we will show shortly, this static graph $\mathcal{G}$ does not correctly capture the true data-generating process, it remains useful as a simple, intuitive summary for presentation. We thus still refer to it, but solely for presentation ease.

**What do the selection variables represent?** Each variable $S^{(t)}$ is a binary indicator of whether an individual in the $t$-th generation reproduces successfully, i.e., produces offspring and thereby contributes their heritable factors $\epsilon^{(t)}$ to the next generation. Note that individual survival is also subsumed by reproduction success. Direct edges from $X^{(t)}$ to $S^{(t)}$ capture how traits affect this outcome, such as how physical

strengths affect mating in animals, or how content and form of ideas affect their circulation in cultural exchange. Finally, an individual exists in generation $T$ only if all preceding generations reproduced successfully, i.e., $S^{(<T)} = 1$.

**But the offspring count does not seem to be represented?** This is because we only focus on data distributions rather than population sizes. While real-world reproduction may yield zero, one, or multiple offspring, our model abstracts away from this, and defines reproduction success as producing exactly *one* offspring: since an individual's multiple offspring can be viewed as i.i.d. draws from $X^{(t+1)}$ given this individual's value of $X^{(t)}$, the offspring count can be distributionally equivalently absorbed as the probability of having one offspring, $p(S^{(t)} = 1 \mid X^{(t)})$, via reweighting.

**Why is there only a singleton $S$ in $\mathcal{G}$?** This is merely for expository clarity. While in static selection settings, it is natural to have multiple selection variables to represent multiple independent selection events, we cannot think of a justification for doing so in the evolutionary setting, where reproduction is the single event to model. This however does not limit generality: the characterization results we establish next still extend when $\mathcal{G}$ has multiple selection variables; please see Appendix B.4 for details.

**Can the causal and selection mechanisms change across generations?** Yes. More specifically, we allow, but do not require these mechanisms to change. Such change is common in real-world scenarios, as fitness often depends on the relative prevalence of traits in the population rather than their exact values. A more striking example is the Manchester moths, where dark coloration was favored during industrial pollution but became disadvantageous once the air was cleaned. Importantly, even if these mechanisms were fixed across generations, the Markov properties we establish next would still hold, as will be formalized in Corollary 1.

**Why do heritable factors operate independently?** This simplification is for the purpose of identifiability, and we do recognize it as a limitation of our model. Think of biological reality: genes can regulate each other, and a trait may also be governed by multiple genes. However, explicitly modeling these complexities would eliminate most CI constraints usable in data, and requires stronger parametric latent-variable models that go beyond the focus of this work. At the same time, this simplification is also standard in literature (Otsuka, 2019), and can be partly justified by noting that $\epsilon^{(T)}$ are not statistically independent anymore once conditioned on $S^{(<T)} = 1$. This aligns with the view that gene dependencies may arise not from regulations, but from coordinated adaptation on the fitness landscape (Bank, 2022).

**Finally, what are the data at hand?** Though we will generalize to heterogeneous settings in §4, for now, data are just considered as i.i.d. samples of one generation, distributed by $p(X^{(T)}|S^{(<T)} = 1)$. No time-sequential or lineage-tracked data are required. Throughout, we assume causal sufficiency w.r.t. $X^{(T)}$, i.e., all variables in $X^{(T)}$ are observed.

Having defined the evolutionary selection model, we now compare it with the static selection model to illustrate why the latter fails to capture crucial information such as CIs in the observed data. Let $\mathcal{G}$ be the static graph and $\mathcal{G}^{(T)}$ the corresponding evolutionary graph, shown in Figure 1. In $\mathcal{G}^{(T)}$, $X_1^{(T)}$ and $X_3^{(T)}$ are d-connected given $X_2^{(T)}$ and $S^{(<T)}$ via an open path through $\epsilon_1^{(T)}$ and $\epsilon_3^{(T)}$, and hence no corresponding CI holds in data. However, this CI is falsely implied by the d-separation $X_1 \perp\!\!\!\perp_d X_3 | X_2, S$ in $\mathcal{G}$. If one were instead to fit a static model to data, false conclusions such as a direct causal relation between $X_1$ and $X_3$ or both being directly selected or directly via $X_2$ could be made.

This discrepancy arises because evolutionary selection does not occur as a one-shot filter, but repeatedly across all preceding generations. Although these earlier generations are unobserved, they leave statistical footprints in the data at the present generation via inheritance. More precisely, evolution induces additional conditional dependencies that are absent under static models, as formalized below:

**Lemma 1 (Additional dependencies induced by evolution).** *Let $\mathcal{G}$ be a DAG over vertices $X \cup \{S\}$, and $\mathcal{G}^{(T)}$ be its corresponding evolutionary graph for some $T \geq 1$. For any disjoint $A, B, C \subseteq X$, if the d-separation $A^{(T)} \perp\!\!\!\perp_d B^{(T)} | C^{(T)}, S^{(<T)}$ holds in $\mathcal{G}^{(T)}$, then $A \perp\!\!\!\perp_d B | C, S$ must also hold in $\mathcal{G}$. However, the converse does not generally hold.*

Lemma 1 shows that evolutionary selection can introduce dependencies in the observed data that, when interpreted improperly via static models, can lead to false discoveries. A careful characterization of these dependencies is therefore essential, and we address this in the next section. The proof of Lemma 1, along with others, are provided in Appendix B.

## 3 Model Characterization and Identification

In this section, we formally characterize the CI constraints in observed data entailed by the model (§3.1), and present a sound and complete model identification procedure that uses these constraints, with particular attention to result interpretation (§3.2). Throughout §3 and §4, we use the graph shown in Figure 2 as a running example for our results.

### 3.1 The Markov Properties

Our ultimate goal is to recover the causal and evolutionary selection mechanisms underlying observed data. Without parametric assumptions on the data-generating process, CI relations become the key source of information. Hence, we now characterize the Markov properties of the evolutionary selection model, i.e., the CI relations it entails in data.

**Lemma 2 (CI implications).** *Let $\mathcal{G}$ be a DAG over vertices*

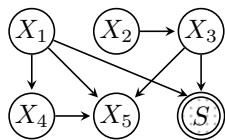

*Figure 2.* A DAG $\mathcal{G}$ used as a running example in §3 and §4. Again, $\mathcal{G}$ corresponds to a static selection model; we use it only for presentation ease, not to represent the true data-generating process. The true evolutionary selection model $\mathcal{G}^{(T)}$ is omitted here.

$X \cup \{S\}$, $\mathcal{G}^{(T)}$ be its corresponding evolutionary graph for some $T \geq 1$, and $A, B, C \subseteq X$ be disjoint sets. Then, the CI relation $A^{(T)} \perp\!\!\!\perp B^{(T)} | C^{(T)}, S^{(<T)} = 1$ holds in all distributions that can be generated by $\mathcal{G}^{(T)}$, if and only if the d-separation $A^{(T)} \perp\!\!\!\perp_d B^{(T)} | C^{(T)}, S^{(<T)}$ holds in $\mathcal{G}^{(T)}$.

Lemma 2 characterizes the CI relations common to all observed distributions that can be generated. A natural question, however, is whether we can always safely assume that these are exactly the CI relations in the specific data at hand, namely, assume *faithfulness*. In particular, recall that in the explanations after Definition 1, we allow causal and selection mechanisms to vary across generations, which is realistic in many settings. But is such variation essential? What if these mechanisms do remain invariant in certain cases?

In more general models, such additional invariance constraints may indeed induce more CIs beyond those implied by d-separations, as seen in symmetric or context-specific models (Højsgaard & Lauritzen, 2008; Boutilier et al., 1996). Fortunately, this concern does not arise here. Owing to the specific structure of the evolutionary selection model, no extra CI relations will be induced. Formally:

**Corollary 1** (**No extra CIs even under additional invariance constraints**). *Let $X, S, \epsilon, \mathcal{G}, \mathcal{G}^{(T)}$ be as in Definition 1. Suppose that in generating data, the selection mechanisms $p(S^{(t)} | X^{(t)})$, causal mechanisms $p(X_i^{(t)} | \mathrm{pa}_{\mathcal{G}^{(T)}}(X_i^{(t)}))$, and inheritance mechanisms $p(\epsilon_i^{(t+1)} | \epsilon_i^{(t)})$ for $X_i \in X$ are all invariant across $t \geq 0$. Then, the CI relations shared by all observed distributions $p(X^{(T)} | S^{(<T)} = 1)$ that can be generated still corresponds to those d-separations in $\mathcal{G}^{(T)}$.*

Corollary 1 justifies assuming *faithfulness*. Any additional CIs would arise only from non-generic, measure-zero parameter cancellations, which we can reasonably disregard.

We may thus proceed purely at the graphical level: Given an evolutionary DAG $\mathcal{G}^{(T)}$, what are the d-separations among observed variables $X^{(T)}$ conditioning on selection variables $S^{(<T)}$? How do they relate to the static graph $\mathcal{G}$ and the generation $T$, and how can they inform graph recovery? At first glance, addressing these questions appears routine: such d-separations induced by conditioning and marginalization in a DAG $\mathcal{G}^{(T)}$ can be characterized by the well-established maximal ancestral graph (MAG) framework (Richardson & Spirtes, 2002), and then be used for graph recovery by the

FCI algorithm (Spirtes et al., 1995; Zhang, 2008).

Surprisingly, however, we find that applying this standard MAG-FCI pipeline is inappropriate here: it not only overcomplicates the graphical analysis but can also lead to incomplete discovery results. This is again owing to the specific structure of the evolutionary graphs, and we defer the detailed explanation to Appendix A. For now, we present a simpler and more informative characterization: instead of using a MAG, the relevant d-separations can be represented directly by a DAG, which we term the *clique-augmented DAG*:

**Definition 2** (**Clique-augmented DAG**). Let $\mathcal{G}$ be a DAG over vertices $X \cup \{S\}$, and $\pi$ an ordering topological to $\mathcal{G}$. The *clique-augmented DAG* of $\mathcal{G}$, denoted by $\mathcal{G}^+$, is a DAG over $X$, where $X_i \to X_j \in \mathcal{G}^+$ if and only if either $X_i \to X_j \in \mathcal{G}$, or $\{X_i, X_j\} \subseteq \mathrm{an}_{\mathcal{G}}(S)$ and $\pi(X_i) < \pi(X_j)$.

An example of such clique-augmented DAGs is shown in Figure 3. The term "clique" refers to the fully connected structure among all variables involved in selection, and "augmented" indicates these additional adjacencies, which, as formalized below, exactly account for the additional dependencies that a static DAG $\mathcal{G}$ fails to capture (Lemma 1), and thus allow this new DAG $\mathcal{G}^+$ to fully represent the CI constraints in data observed under evolutionary selection:

**Theorem 1** (**Clique-augmented DAG representation for evolutionary selection models**). *Let $X, S, \mathcal{G}, \mathcal{G}^{(T)}, \mathcal{G}^+$ be as in Definitions 1 and 2. For any $T \geq 1$ and any disjoint sets $A, B, C \subseteq X$, the d-separation $A^{(T)} \perp\!\!\!\perp_d B^{(T)} | C^{(T)}, S^{(<T)}$ holds in the DAG $\mathcal{G}^{(T)}$, if and only if the d-separation $A \perp\!\!\!\perp_d B | C$ holds in the DAG $\mathcal{G}^+$.*

Theorem 1 has the following three important implications:

First, although the distributions $p(X^{(T)} | S^{(<T)} = 1)$ may change with $T$, and may even never converge, their entailed CI relations remain invariant. Hence in practice, one needs not to know the exact generation $T$ of the observed data, or to assume that the evolution has reached any equilibrium.

Second, when reproduction happens completely at random, i.e., $\mathrm{pa}_{\mathcal{G}}(S) = \varnothing$, the DAG $\mathcal{G}^+$ reduces to the DAG $\mathcal{G}$ (with $S$ removed). In this degenerated case, the static graphs suffice to capture CI relations in data even under evolution.

Third, evolutionary selection can only be falsified but not confirmed from data. In particular, one can never conclude that a variable $X_i$ *is* involved in selection (i.e., $X_i \in \mathrm{an}_{\mathcal{G}}(S)$), but only that it *is not*, because the observable CI constraints can always be equivalently represented by the DAG $\mathcal{G}^+$ where no selection occurs. This is fundamentally different from the static selection setting, where it is sometimes possible to confirm a variable's involvement in selection, or even to determine whether it is directly selected.

These implications, especially the third one, are essential for the model identification procedure, as presented next.

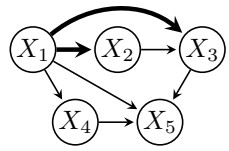 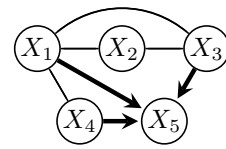

*Figure 3.* An illustration of model identification with single-domain data (§3). Let $\mathcal{G}$ be the DAG in Figure 2. **Left:** The *clique-augmented DAG* $\mathcal{G}^+$ from Definition 2; highlighted are the augmented edges to form a clique among $\mathrm{an}_{\mathcal{G}}(S) = \{X_1, X_2, X_3\}$. **Right:** CPDAG $\mathcal{C}$ output by Algorithm 1, which is also the CPDAG of $\mathcal{G}^+$. One may verify Theorem 2 on the highlighted oriented edges: indeed, $\{X_1, X_3, X_4\} \subseteq \mathrm{pa}_{\mathcal{G}}(X_5)$, and $X_5 \notin \mathrm{an}_{\mathcal{G}}(S)$.

### 3.2  Model Identification

In §3.1, we characterized the Markov properties of the evolutionary selection model, establishing how CI relations in the observed data correspond to d-separations in the graph and how they, in turn, are represented by the clique-augmented DAG. Now, we present how these CI relations can be used to identify the model, up to its identifiability limits.

Thanks to the representability of all CIs in data via a clique-augmented DAG without latent or selection variables, the identification algorithm reduces to a standard form: one may treat data *as if* no selection or evolution were ever present:

**Algorithm 1** (**Model identification from single-domain data**). **Input:** An observed data matrix $X \in \mathbb{R}^{n \times d}$. **Procedure:** Apply PC or GES (Spirtes et al., 2000; Chickering, 2002), or any other nonparametric method that is sound and complete under causal sufficiency and faithfulness assumptions, to data $X$. **Output:** A completed partially directed acyclic graph (CPDAG) over the $d$ vertices indexed by $X$.

At this point, one might find it surprising or even paradoxical: after detailing all the complexities introduced by selection and evolution, are we now suggesting that they can be totally ignored, and a standard PC algorithm suffices? The answer is no: these complexities do not arise in the algorithmic construction of the CPDAG in Algorithm 1, but in its *interpretation*, which must be carefully distinguished from the standard interpretations in settings where these algorithms were originally intended for, or where selection is static.

We formalize this interpretation as follows:

**Theorem 2** (**Soundness and completeness of Algorithm 1**). *Suppose the observed data are large enough i.i.d. samples drawn from $p(X^{(T)}|S^{(<T)} = 1)$, with $X$, $S$, $\mathcal{G}$, $\mathcal{G}^{(T)}$ be as in Definition 1, and assume faithfulness w.r.t. Lemma 2. Let $\mathcal{C}$ be the CPDAG output by Algorithm 1 on data. Then:*

***Soundness and completeness of adjacencies:*** *Two vertices $X_i$ and $X_j$ are adjacent in $\mathcal{C}$, if and only if they are directly causally related, or both involve in selection; i.e., $X_i \in \mathrm{pa}_{\mathcal{G}}(X_j)$, or $X_i \in \mathrm{ch}_{\mathcal{G}}(X_j)$, or $\{X_i, X_j\} \subseteq \mathrm{an}_{\mathcal{G}}(S)$.*

***Soundness of orientations:*** *For any oriented edge $X_i \to$*

$X_j \in \mathcal{C}$, *this direct causal relation is true, and $X_j$ is not involved in selection, i.e., $X_i \in \mathrm{pa}_{\mathcal{G}}(X_j)$ and $X_j \notin \mathrm{an}_{\mathcal{G}}(S)$.*

***Completeness of orientations:*** *For any unoriented edge $X_i—X_j \in \mathcal{C}$, no further identification is possible; i.e., there always exists an alternative DAG $\mathcal{G}'$ that differs from $\mathcal{G}$ in the relation between $X_i, X_j$ (among the three possibilities for adjacency), yet induces the same CPDAG output $\mathcal{C}$.*

One may use Theorem 2 to interpret the example output of Algorithm 1 shown in Figure 3. Theorem 2 highlights the additional complexities introduced by evolution: when selection is static, roughly speaking, the "spurious" dependencies only occur among variables that are directly selected, while evolution propagates such dependencies even to those indirect ones. Moreover, unlike in the static setting, now the presence of selection can no longer be theoretically confirmed from data alone, though in practice, when a clique is detected in which pairwise direct causal relations are implausible (e.g., based on prior knowledge), their joint selection involvement can be a compelling alternative explanation.

To conclude this section, we reflect on three representative scenarios to illustrate how improper handling of data generated under evolutionary selection, in contrast to our approach, can lead to unsound or incomplete discovery results.

**Scenario 1 (Ignoring selection).** If selection is ignored entirely, one may apply PC/GES and obtain a CPDAG identical to the output of Algorithm 1. However, one would incorrectly interpret all adjacencies as direct causal relations.

**Scenario 2 (Modeling selection as static).** If selection is acknowledged but treated as static, one may apply FCI and obtain a partial ancestral graph (PAG). However, false interpretations of adjacencies can be made by missing the possibility of joint indirect selection involvement. Moreover, one may expect certain edge types in PAG to signal selection, which would never appear. See Appendix A for details.

**Scenario 3: (Correctly modeling evolution but defaulting to MAG–FCI pipeline).** When selection is acknowledged and even evolution is correctly modeled as in Definition 1, directly applying the standard MAG–FCI pipeline is still incomplete. Certain structures that are in fact identifiable would still be treated as ambiguous in the output PAG. This is because the specific structure of evolutionary graphs imposes additional constraints. See Appendix A for details.

These scenarios altogether emphasize a key lesson: under evolution, algorithm outputs require careful interpretation; they may differ substantially from their standard meaning.

## 4  Generalization to Heterogeneous Data

In §3, we showed how the underlying data-generating process can be identified from i.i.d. observations of one generation in one environment. In this section, we generalize this

framework to heterogeneous data, such as observations of multiple generations in the same environment (e.g., Manchester moths during and after the Industrial Revolution) and of different environments (e.g., moths in Manchester versus Sydney). Such data are also commonly available in real world and often reflect changes in causal and selection mechanisms underlying evolution. We show how these changes can be used to further improve model identifiability.

We first define what we mean by *changes*. We term different generations or environments uniformly as different *domains*.

**Definition 3** (**Change of mechanisms across domains**). Let $X, S, \mathcal{G}$ be as in Definition 1. Consider multiple domains parameterized by $p^{(1)}, \ldots p^{(K)}$ from their corresponding evolutionary graphs $\mathcal{G}^{(T_1)}, \ldots, \mathcal{G}^{(T_K)}$. We say the selection mechanism is *changed*, or simply $S$ is changed, when there exists $k_1 \neq k_2$, s.t. $p^{(k_1)}(S^{(t)}|X^{(t)}) \neq p^{(k_2)}(S^{(t)}|X^{(t)})$ for some $t < \min(T_{k_1}, T_{k_2})$. We say the causal mechanism of an $X_i \in X$ is *changed*, or simply $X_i$ is changed, when there exists $k_1 \neq k_2$, s.t. $p^{(k_1)}(X_i^{(t_1)}|\mathrm{pa}_{\mathcal{G}^{(T_{k_1})}}(X_i^{(t_1)})) \neq p^{(k_2)}(X_i^{(t_2)}|\mathrm{pa}_{\mathcal{G}^{(T_{k_2})}}(X_i^{(t_2)}))$ for $t_1, t_2 = T_{k_1}, T_{k_2}$ and for at least another pair of $t_1 = t_2 < \min(T_{k_1}, T_{k_2})$. Finally, we denote the set of changed variables by $I \subseteq X \cup \{S\}$. Note that we assume the causal and selection structures remain fixed, while only their parameterizations can change.

We then show how such changes induce testable implications in multi-domain observed data. As CIs in each domain already characterized, we focus on another information: the (in)variance of conditional distributions across domains. Let $\zeta$ be the domain index; these invariances can be expressed as CIs between $\zeta$ and other variables in $X$. Previous works show that, in standard settings, these CIs correspond exactly to d-separations in the causal graph with $\zeta$ as an additional root and pointing to changed variables (Zhang et al., 2015; Huang et al., 2020; Mooij et al., 2020). We show that a similar characterization also holds in our setting, though constructing this graph requires additional care:

**Theorem 3** (**Multi-domain clique-augmented DAG representation**). *Let all notations be as in Definitions 1 to 3. We construct a new DAG $\mathcal{G}^{+I}$ by adding an auxiliary vertex $\zeta$ to $\mathcal{G}^+$ and draw an edge $\zeta \to X_i$ for each $X_i \in I$. Further, if $\mathrm{an}_{\mathcal{G}}(S) \cap I \neq \varnothing$, that is, when selection itself or any variable involved in selection is changed, we add edges from $\zeta$ to all of $\mathrm{an}_{\mathcal{G}}(S)\backslash\{S\}$ to $\mathcal{G}^{+I}$. Then, for any sets $A, C \subseteq X$, a d-separation $\zeta \perp\!\!\!\perp_d X_A \mid X_C$ in $\mathcal{G}^{+I}$ implies invariance of the observed conditional distribution $p^{(k)}(X_A^{(T_k)}|X_C^{(T_k)}, S^{(<T_k)} = 1)$ across all the $K$ domains.*

We name the DAG $\mathcal{G}^{+I}$ over vertices $X \cup \{\zeta\}$ as the *multi-domain clique-augmented DAG*. Note that an edge $\zeta \to X_i \in \mathcal{G}^{+I}$ does not imply that $X_i$ itself is changed; it may be due to selection change, as shown in Figure 4. Now, just as how the DAG representation $\mathcal{G}^+$ suggests the use of PC

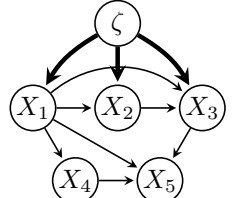 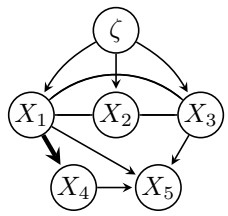

*Figure 4.* An illustration of model identification with multi-domain data (§4). Let $\mathcal{G}, \mathcal{G}^+, \mathcal{C}$ be as in Figures 2 and 3. Let $I = \{S\}$, i.e., only the selection mechanism changes across domains. **Left:** The *multi-domain clique-augmented DAG* $\mathcal{G}^{+I}$ from Theorem 3. Comparing $\mathcal{G}^{+I}$ against $\mathcal{G}^+$, highlighted are the additional edges from $\zeta$, but note that the causal mechanisms of $X_1, X_2, X_3$ themselves in fact do not change. **Right:** PDAG $\mathcal{P}$ output by Algorithm 2. Comparing $\mathcal{P}$ against $\mathcal{C}$, the newly oriented edge $X_1 \to X_4$ (highlighted) illustrates how heterogeneity helps improve identifiability. Verify Theorem 4: indeed, $X_1 \in \mathrm{pa}_{\mathcal{G}}(X_4)$, and $X_4 \notin \mathrm{an}_{\mathcal{G}}(S)$.

in §3, this DAG $\mathcal{G}^{+I}$ suggests that we can, again, directly apply the existing standard multi-domain methods, even though they were not intended for evolutionary selection:

**Algorithm 2** (**Model identification from multi-domain data**). **Input:** Observed data from multiple domains over a common variable set $X$. **Procedure:** Apply CDNOD (Huang et al., 2020) or comparable methods. **Output:** A partially directed acyclic graph (PDAG) over vertices $X \cup \{\zeta\}$.

Comparing to Algorithm 1, Algorithm 2 orient more edges (and all ones identifiable) given the observed changes:

**Theorem 4** (**Soundness and completeness of Algorithm 2**). *Let all notations be as in Definitions 1 to 3. Assume faithfulness w.r.t. Lemma 2 and Theorem 3. Let $\mathcal{P}_X$ be the subgraph of Algorithm 2's output PDAG on $X$. Then, all statements of Theorem 2 continue to hold when '$\mathcal{C}$' is replaced by '$\mathcal{P}_X$'; $X_i \to X_j \in \mathcal{C}$ implies $X_i \to X_j \in \mathcal{P}_X$, but not vice versa.*

An example of the PDAG output of Algorithm 2 is shown in Figure 4. We conclude this section with two remarks:

First, the presence of selection still cannot be confirmed, even with data heterogeneity that enables identification of more causal relations and non-involvements in selection. This is reflected in the absence of $S$ from the DAG $\mathcal{G}^{+I}$.

Second, suppose selection exists and changes (e.g., $I = \{S\}$). Then in the DAG $\mathcal{G}^{+I}$, all variables involved in selection are directly "caused" by $\zeta$, though their own causal mechanisms are unchanged. This aligns with our common understanding: when we say "environment and evolution influence traits," this influence may often act through changes in fitness preferences across environments and during evolution, rather than by directly altering traits themselves.

## 5 Experiments and Results

In this section, we present empirical studies on synthetic and real-world data to demonstrate that our method effectively recovers causal mechanisms under evolutionary selection.

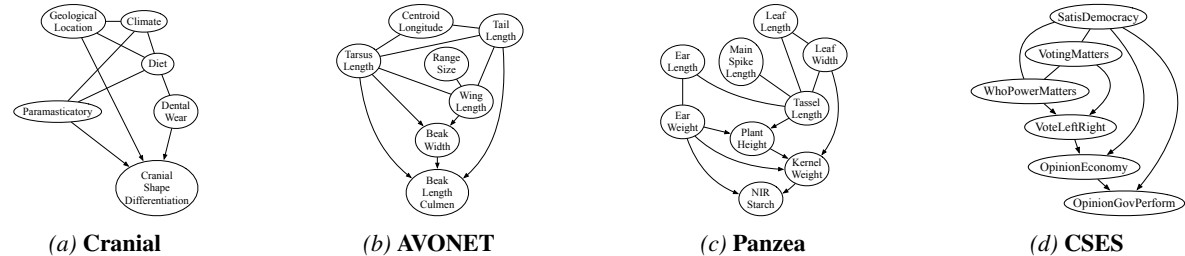

*(a)* **Cranial**          *(b)* **AVONET**          *(c)* **Panzea**          *(d)* **CSES**

*Figure 5.* Subgraphs of learned CPDAGs/PDAGs on four datasets. Oriented edges (→) indicate causal relations whose effect variables are not involved in selection. Selection, if present, must be among cliques linked by unoriented edges (—). Full results in Appendix D.2.

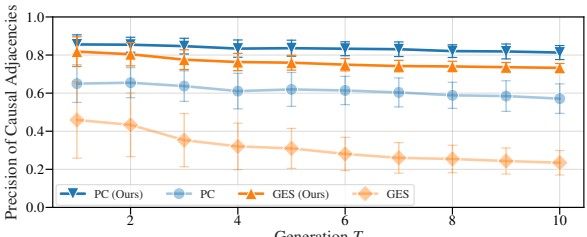

*Figure 6.* Precision (y-axis) of causal adjacencies for $d = 20$ observed variables, using the standard interpretation and ours. Mean and standard deviation over 50 random runs are shown.

### 5.1   Experiments on Synthetic Data

We evaluate our method on synthetic data generated according to the evolutionary selection model in Definition 1. Specifically, to simulate causal mechanisms within generations, we first generate a random Erdős–Rényi graph (Erdős & Rényi, 1959) among $d \in \{10, 15, 20\}$ observable variables $X$ with an average degree of 2. Next, to simulate preferential fitness, we add a selection variable $S$ with $d/5$ parents randomly chosen from $X$. A linear structural equation model (SEM) is then instantiated on this graph, with edge coefficients drawn uniformly from $[-2, -0.5] \cup [0.5, 2]$ and variances of Gaussian noise terms $\epsilon$ from $[1, 4]$. Unlike static selection where samples are filtered based on $S$, here each sample in generation $t$ produces between 0 and 5 offspring, determined by the percentile rank of $S^{(t)}$ within generation $t$, reflecting the relative fitness on reproduction. Each offspring inherits its noise term $\epsilon^{(t+1)}$ from $\epsilon^{(t)}$ plus a unit-variance Gaussian perturbation, and generates its $X^{(t+1)}$ according to the same SEM. Repeating this process for $T$ generations yields a population evolved under selection.

Our goal is to validate Theorem 2, that is, only the oriented edges in the output CPDAGs by standard algorithms are guaranteed to be true causal relations. We apply PC (Spirtes et al., 2000) and GES (Chickering, 2002) to the data and compare our interpretation, a more conservative one, to the standard interpretation that treats all adjacencies as causal relations. We report the precision of each interpretation in recovering true causal adjacencies, regardless of directions.

The results for $d = 20$ are presented in Figure 6. Notably, our interpretation consistently achieves higher precision than the standard one for both PC and GES, and remains sta-

ble across generations $T$. The lower precision for standard GES is partly from its tendency to output denser graphs with many unoriented edges, though its oriented edges remain reliable. Results for $d = 10, 15$ are provided in Appendix D.1.

We also compare with another method for latent-variable causal discovery, and conduct experiments under model misspecifications with i) latent confounding, ii) dependent inheritance, and iii) structural changes across domains (edge reversals). See also Appendix D.1 for these further results.

### 5.2   Experiments on Real-World Data

We evaluate our methods on seven real-world datasets, including *Drosophila* gene expressions (**DGRP**; Everett et al. (2020)), archaeological human cranial measurements (**Cranial**; Noback et al. (2011)), agronomic morphology data for maize (**Panzea**; Zhao et al. (2006)), phenotypic trait data for mammals (**PanTHERIA**; Jones et al. (2009)) and birds (**AVONET**; Tobias et al. (2022)), cross-national political survey data from the Comparative Study of Electoral Systems (**CSES**; link), and demographic microdata from the U.S. Census American Community Survey (**PUMS**; link).

Figure 5 shows four example subgraphs of learned CPDAGs/PDAGs (depending on whether multi-domain data are used) over key variable sets. Among them, many inferred relations are readily interpretable and align with domain knowledge. In the **Cranial** data, geography, climate, and dental wear causally affect cranial shape, which itself does not appear to be involved in fitness selection; selection, if any, likely operates upstream, such as among climate and diet (see the unoriented cliques). Similarly, in **AVONET**, beak morphology is also downstream, with selection, if any, likely acting on locomotion traits (e.g., wing and tarsus) that influence ecological access and shape dietary (latent), and thus beak-variation. In **Panzea**, the causal ordering from vegetative and reproductive traits to yield-related outcomes seems reasonable. Finally, in **CSES**, the clique among attitudinal variables (e.g., beliefs about voting efficacy) may suggest differential propagation of views.

Note that the analyses above are qualitative. For quantitative evaluation, we tried to find evolutionary-selection-related datasets with ground-truth causal graphs, but were unable

to find any. That said, to make the evaluation more convincing, we have added the following analyses: For the DGRP dataset, we use the reported eQTL-probed (which may be incomplete) regulator–target pairs as partial ground truth. For the rest six datasets, we use the LLM-generated pseudo ground truth. Full results are provided in Appendix D.2.

## 6 Conclusion and Limitations

In this work, we introduce the evolutionary selection model to represent data generated with differential fitness in evolution, a setting often ignored or conflated with static selection. We then show how underlying mechanisms can still be reliably recovered from such data. A key limitation is that heritable factors are assumed to operate independently, motivating future work on parameterized and equilibrium-aware models for more complex inheritance scenarios. Other limitations include the model's specificity to the evolution use case, and the assumptions of causal sufficiency, faithfulness, and cross-domain structural invariance.

## Acknowledgment

We would like to acknowledge the support from NSF Award No. 2229881, AI Institute for Societal Decision Making (AI-SDM), the National Institutes of Health (NIH) under Contract R01HL159805, and grants from Quris AI, Florin Court Capital, MBZUAI-WIS Joint Program, and the Al Deira Causal Education project. We also thank the anonymous reviewers for their helpful suggestions.

## Impact Statement

This paper presents work whose goal is to advance the field of causal discovery. Our method has implications for analyzing e.g., demographical traits, political attitudes, and cultural differences in social science related datasets. While our work enhances such tasks, responsible application is crucial to avoid misinterpretation in the results.

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

# A  More Elaborations on Motivations

We first elaborate on more about our model choice: why we directly uses the augmented DAGs to represent the d-separations in the model, instead of using the common MAG-FCI pipeline. At the same time, we also establish the alternative MAG representation, which, though not directly used for model identification, plays an essential role in proving the results to be given in the next section.

We now introduce the MAG basics. A MAG is a mixed graph with three possible kinds of edges: directed ($\rightarrow$), bi-directed ($\leftrightarrow$), and undirected (—). Given a DAG over vertices partitioned by $O$ (observed), $L$ (latent), and $S$ (selected), a MAG over vertices $O$ can be uniquely constructed. Here we omit the specific construction rules and focus on its purpose: to capture the d-separations among $O$ marginalized over $L$ and conditioned on $S$. In particular, $O_i$ and $O_j$ are adjacent in the MAG if and only if they cannot be d-separated by any remaining variables in $O$ together with $S$ in the original DAG. Edge orientations capture finer-grained properties, which can be referred to (Richardson & Spirtes, 2002).

With the MAGs, the complex structure of our evolutionary models can also be abstracted in the following way:

**Theorem A.1** (**MAG representation for evolutionary selection models**). *Let $X$, $S$, $\mathcal{G}$, $\mathcal{G}^{(T)}$ be as in Definition 1. The MAG induced by $\mathcal{G}^{(T)}$ over $X^{(T)}$ is structurally invariant across $T \geq 1$, up to the $T$-indices. For notation ease, we thus omit the $T$-index and denote this MAG by $\mathcal{E}(\mathcal{G})$ over vertices $X$, which we refer to as the "Evolutionary MAG". $\mathcal{E}(\mathcal{G})$ has the following adjacencies and orientations:*

- *Two vertices $X_i$ and $X_j$ are adjacent in $\mathcal{E}(\mathcal{G})$ if and only if they are adjacent in $\mathcal{G}$, or $\{X_i, X_j\} \subseteq \mathrm{an}_{\mathcal{G}}(S)$.*
- *An edge between $X_i$ and $X_j$ is oriented as $X_i \rightarrow X_j$ if $X_i \in \mathrm{an}_{\mathcal{G}}(X_j)$, as $X_j \rightarrow X_i$ if $X_j \in \mathrm{an}_{\mathcal{G}}(X_i)$, and as $X_i \leftrightarrow X_j$ otherwise. There are no undirected edges.*

The proof of Theorem A.1 is provided in Appendix B, and it is essential in proving our DAG representation (Theorem 1).

We need to emphasize that $\mathcal{E}(\mathcal{G})$ denotes the "Evolutionary MAG" of $\mathcal{G}^{(T)}$. It is *not* the MAG of $\mathcal{G}$, which we distinguish by the notation $\mathcal{S}(\mathcal{G})$, termed "Static MAG". These two are fundamentally different, as illustrated in Figure 7:

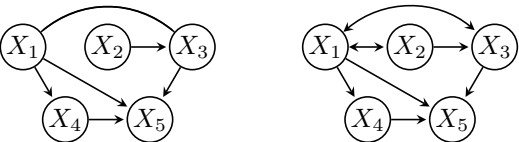

*Figure 7.* An illustration of the static MAG and evolutionary MAG. Let $\mathcal{G}$ be the DAG as shown in Figure 2. Left is the MAG $\mathcal{E}(\mathcal{G})$ MAG on $\mathcal{G}$ by directly conditioning on the $S$ variable, while right is the $\mathcal{S}(\mathcal{G})$ MAG over DAG $\mathcal{G}^{(T)}$ by conditioning on the $S^{(<T)}$ and marginalizing out the remaining latent variables. In terms of adjacencies, the right MAG has one more adjacency between $X_1$ and $X_2$: in the original DAG $\mathcal{G}$, even with $S$ conditioned on, $X_1$ and $X_2$ can still be d-separated by $X_3$. This is however not the case for the evolutionary MAGs.

Just like CPDAG that is used to represent an equivalence class of DAGs, the PAG (partial ancestral graph) is used to represent an equivalence class of MAGs. A PAG thus has 6 types of edges: $\rightarrow$, $\leftrightarrow$, —, $\circ$—, $\circ\rightarrow$, and $\circ$—$\circ$, where the edge mark of '$\circ$' indicates multiple possibilities (variant) edge marks across all MAGs in the equivalence class, and other edge marks indicate the fixed marks. If we directly use the FCI on the running example, the output PAG is presented as in Figure 8:

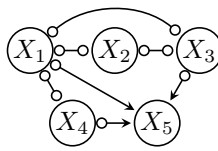

*Figure 8.* An illustration of the output PAG on the running example of Figure 2. One may also use it to compare against the MAG $\mathcal{E}(\mathcal{G})$ shown in Figure 7.

Now, if one were to directly use the MAG-FCI pipeline and try to interpret the edge marks according to the MAG construction rule as in Theorem A.1, they will get uninformative results. Specifically, for the $X_i \circ\rightarrow X_j$ edges, they may falsely consider into the possibility of $X_i \leftrightarrow X_j$, i.e., the two may be both involved in selection and not directly causally related. Even if they try to interpret this edge as $X_i \rightarrow X_j$, the MAG construction rule in Theorem A.1 only suggests an ancestral relation

between the two; they may not directly reach the conclusion that this edge must be a direct causal relation. Further, during the running process of FCI, much unnecessary effort would be put on finding e.g., discriminating paths and other orientation rules, which would not happen in this case.

Therefore, we get motivation from the MAG construction, but in the paper we directly use the augmented DAG representation. This is because the clique structure among $\mathrm{an}_\mathcal{G}(S)$ induces that no v-structures (in the MAG) exists, and no edges among them can be oriented. Thus just a DAG with same adjacencies in this clique would also suffice the represent all the corresponding d-separations.

# B  Proofs of Main Results

## B.1  Proof of Lemma 1

**Lemma 1 (Additional dependencies induced by evolution).** *Let $\mathcal{G}$ be a DAG over vertices $X \cup \{S\}$, and $\mathcal{G}^{(T)}$ be its corresponding evolutionary graph for some $T \geq 1$. For any disjoint $A, B, C \subseteq X$, if the d-separation $A^{(T)} \perp\!\!\!\perp_d B^{(T)} | C^{(T)}, S^{(<T)}$ holds in $\mathcal{G}^{(T)}$, then $A \perp\!\!\!\perp_d B | C, S$ must also hold in $\mathcal{G}$. However, the converse does not generally hold.*

*Proof of Lemma 1.* Let $\mathcal{G}$ be the static DAG and $\mathcal{G}^{(T)}$ be the evolutionary graph for generation $T$. We prove by contrapositive. For any d-connection $A \not\perp\!\!\!\perp_d B | C, S$ in $\mathcal{G}$, consider an open path $p$ between $A$ and $B$ in $\mathcal{G}$, and we can construct another open path between $A^{(T)}$ and $B^{(T)}$ in $\mathcal{G}^{(T)}$. Specifically, for each collider $W$ on $p$, 1) if $W$ is not an ancestor of $S$ in $\mathcal{G}$, then $W$ must be an ancestor of some $c \in C$ (so as $W^{(T)}$ and $c^{(T)}$), and we can keep the path going among $X^{(T)}$ in $\mathcal{G}^{(T)}$; 2) Otherwise, $W$ is an ancestor of $S$, and consider the nearest "trek top" nodes on the sides of $W$, as $\cdots \leftarrow X_i \rightarrow \cdots \rightarrow W \leftarrow \cdots \leftarrow X_j \rightarrow \cdots$ on $p$ (note that $X_i, X_j$ can be $A, B$ themselves). We can reroute the path through previous time slices, by $\cdots \leftarrow X_i^{(T)} \leftarrow \epsilon_i^{(T)} \leftarrow \epsilon_i^{(T-1)} \rightarrow X_i^{(T-1)} \rightarrow \cdots \rightarrow W^{(T-1)} \leftarrow \cdots \leftarrow X_j^{(T-1)} \leftarrow \epsilon_j^{(T-1)} \rightarrow \epsilon_j^{(T)} \rightarrow X_j^{(T)} \rightarrow \cdots$. Since the collider $W^{(T-1)}$ is an ancestor of $S^{(T-1)}$ and $S^{(T-1)}$ is conditioned on, this path remains open. By this construction, we get the open path in $\mathcal{G}^{(T)}$, and thus $A^{(T)} \not\perp\!\!\!\perp_d B^{(T)} | C^{(T)}, S^{(<T)}$ still holds in $\mathcal{G}^{(T)}$.

Note that the converse does not generally hold (i.e., d-separation in static graph does not imply d-separation in evolutionary graph), by just considering the counterexample in Figure 1. $\qquad\square$

## B.2  Proof of Lemma 2

**Lemma 2 (CI implications).** *Let $\mathcal{G}$ be a DAG over vertices $X \cup \{S\}$, $\mathcal{G}^{(T)}$ be its corresponding evolutionary graph for some $T \geq 1$, and $A, B, C \subseteq X$ be disjoint sets. Then, the CI relation $A^{(T)} \perp\!\!\!\perp B^{(T)} | C^{(T)}, S^{(<T)} = 1$ holds in all distributions that can be generated by $\mathcal{G}^{(T)}$, if and only if the d-separation $A^{(T)} \perp\!\!\!\perp_d B^{(T)} | C^{(T)}, S^{(<T)}$ holds in $\mathcal{G}^{(T)}$.*

Proof of Lemma 2 follows directly from global Markov properties, and we omit it here. While the proof is simple, we include Lemma 2 mainly as a reference point in the overall logic. As a bridge from CI relations in the data to d-separations in the graph, it allows us to next i) justify faithfulness (Corollary 1), and ii) further move to a simpler DAG representation (Theorem 1).

## B.3  Proof of Corollary 1

**Corollary 1 (No extra CIs even under additional invariance constraints).** *Let $X$, $S$, $\epsilon$, $\mathcal{G}$, $\mathcal{G}^{(T)}$ be as in Definition 1. Suppose that in generating data, the selection mechanisms $p(S^{(t)}|X^{(t)})$, causal mechanisms $p(X_i^{(t)} | \mathrm{pa}_{\mathcal{G}^{(T)}}(X_i^{(t)}))$, and inheritance mechanisms $p(\epsilon_i^{(t+1)}|\epsilon_i^{(t)})$ for $X_i \in X$ are all invariant across $t \geq 0$. Then, the CI relations shared by all observed distributions $p(X^{(T)}|S^{(<T)} = 1)$ that can be generated still corresponds to those d-separations in $\mathcal{G}^{(T)}$.*

*Proof of Corollary 1.* The assumption of invariant mechanisms implies that the parameters of the structural equations (or conditional probability tables) are tied across time steps $t$. While parameter tying can essentially create cancellations in specific models (e.g., linear Gaussian symmetric models), it generally does not induce new conditional independencies in general nonlinear/nonparametric models, as long as there are no deterministic relations. This is indeed the case in our setting, where the inheritance edges $\epsilon^{(t)} \rightarrow \epsilon^{(t+1)}$ are with randomness. Suppose, however that $\epsilon^{(t)}$ are passed without change, then conditioning certain values of $X^{(T)}$ would happen to be conditioning the previous generations $X^{(<T)}$, breaking the CI implications. As long as the inheritance edges are with additional noise, no other deterministic relations can arise. And this

is exactly the case in our model, where no other cases share a same set of variables (or its copy when value is the same) as parent input. □

## B.4   Proof of Theorem 1

**Theorem 1 (Clique-augmented DAG representation for evolutionary selection models).** *Let $X$, $S$, $\mathcal{G}$, $\mathcal{G}^{(T)}$, $\mathcal{G}^+$ be as in Definitions 1 and 2. For any $T \geq 1$ and any disjoint sets $A, B, C \subseteq X$, the d-separation $A^{(T)} \perp\!\!\!\perp_d B^{(T)} \mid C^{(T)}, S^{(<T)}$ holds in the DAG $\mathcal{G}^{(T)}$, if and only if the d-separation $A \perp\!\!\!\perp_d B \mid C$ holds in the DAG $\mathcal{G}^+$.*

*Proof of Theorem 1.* To prove the DAG representation (Theorem 1), we first prove the MAG representation (Theorem A.1):

**Theorem A.1 (MAG representation for evolutionary selection models).** *Let $X$, $S$, $\mathcal{G}$, $\mathcal{G}^{(T)}$ be as in Definition 1. The MAG induced by $\mathcal{G}^{(T)}$ over $X^{(T)}$ is structurally invariant across $T \geq 1$, up to the $T$-indices. For notation ease, we thus omit the $T$-index and denote this MAG by $\mathcal{E}(\mathcal{G})$ over vertices $X$, which we refer to as the "Evolutionary MAG". $\mathcal{E}(\mathcal{G})$ has the following adjacencies and orientations:*

- *Two vertices $X_i$ and $X_j$ are adjacent in $\mathcal{E}(\mathcal{G})$ if and only if they are adjacent in $\mathcal{G}$, or $\{X_i, X_j\} \subseteq \mathrm{an}_{\mathcal{G}}(S)$.*
- *An edge between $X_i$ and $X_j$ is oriented as $X_i \rightarrow X_j$ if $X_i \in \mathrm{an}_{\mathcal{G}}(X_j)$, as $X_j \rightarrow X_i$ if $X_j \in \mathrm{an}_{\mathcal{G}}(X_i)$, and as $X_i \leftrightarrow X_j$ otherwise. There are no undirected edges.*

*Proof of Theorem A.1.* Fix $T \geq 1$. View $\mathcal{G}^{(T)}$ as a DAG whose vertex set can be partitioned as $V(\mathcal{G}^{(T)}) = O \cup L \cup \tilde{S}$, $O := X^{(T)}$, $\tilde{S} := S^{(<T)}$, $L := V(\mathcal{G}^{(T)}) \setminus (O \cup \tilde{S})$, so that $O$ are the observed vertices, $\tilde{S}$ are the conditioned (selection) vertices, and $L$ are latent/unobserved. Let $\mathrm{MAG}(\mathcal{G}^{(T)}; O, \tilde{S})$ denote the maximal ancestral graph obtained from $\mathcal{G}^{(T)}$ by marginalization over $O$ while conditioning on $\tilde{S}$ (as in Richardson & Spirtes (2002)); we only use the standard facts that (i) adjacencies correspond to the existence of an inducing path, and (ii) orientations follow the projection rules and reflect ancestral relations).

**Step 1: Structural invariance in $T$.** The only edges in $\mathcal{G}^{(T)}$ that connect different generations are the inheritance edges $\epsilon_i^{(t)} \rightarrow \epsilon_i^{(t+1)}$ and $\epsilon_i^{(t)} \rightarrow X_i^{(t)}$ (and the within-generation causal edges among the $X^{(t)}$ mirroring $\mathcal{G}$). Crucially, $\mathcal{G}^{(T)}$ is time-homogeneous in structure: every generation contains the same copy of $\mathcal{G}$ over $X^{(t)} \cup \{S^{(t)}\}$ and the same inheritance wiring via the $\epsilon^{(t)}$'s. Therefore, any path-based criterion among $X^{(T)}$ after conditioning on $S^{(<T)}$ depends only on the static ancestral relations in $\mathcal{G}$ (more specifically, only need the graph copy at $T - 1$; and not on the numerical value of $T$), except for the superscripts. Hence $\mathrm{MAG}(\mathcal{G}^{(T)}; X^{(T)}, S^{(<T)})$ is identical across $T$ up to relabeling $X_i^{(T)} \mapsto X_i$. We denote this common graph by $\mathcal{E}(\mathcal{G})$.

**Step 2: Adjacency characterization via inducing paths.** Recall (e.g., Richardson & Spirtes, 2002) that two vertices $u, v \in O$ are adjacent in the induced MAG if and only if there exists an *inducing path* between $u$ and $v$ in the original DAG relative to $(O, \tilde{S})$: a path whose non-endpoints are in $L \cup \tilde{S}$, every collider on the path is an ancestor of $\{u, v\} \cup \tilde{S}$, and every non-collider lies in $L$ (so it is marginalized out).

Fix distinct $i, j$.

($\Rightarrow$) *If $X_i$ and $X_j$ are adjacent in $\mathcal{G}$, or $\{X_i, X_j\} \subseteq \mathrm{an}_{\mathcal{G}}(S)$, then $X_i$ and $X_j$ are adjacent in $\mathcal{E}(\mathcal{G})$.*

*Case 1: $X_i$ and $X_j$ are adjacent in $\mathcal{G}$.* Trivial, since $X_i^{(T)}$ and $X_j^{(T)}$ are adjacent in the MAG.

*Case 2: $\{X_i, X_j\} \subseteq \mathrm{an}_{\mathcal{G}}(S)$.* By definition of ancestor, there must exists a path $X_i \rightarrow \cdots \rightarrow K \leftarrow \cdots \leftarrow X_j$ in the $\mathcal{G}$, where $K$ is the only collider on the path and $K$ is also an ancestor of $S$ (note that $K$ can be $S$ itself). Then, simply lift this path to some generation $t < T$ ($T - 1$ suffices) in $\mathcal{G}^{(T)}$ to connect $X_i^{(T)}$ back to $X_i^{(t)}$ and $X_j^{(t)}$ forward to $X_j^{(T)}$: $X_i^{(T)} \leftarrow \epsilon_i^{(T)} \leftarrow \cdots \leftarrow \epsilon_i^{(t)} \rightarrow X_i^{(t)} \rightarrow \cdots \rightarrow X_k^{(t)} \leftarrow \cdots \leftarrow X_j^{(t)} \leftarrow \epsilon_j^{(t)} \rightarrow \cdots \rightarrow \epsilon_j^{(T)} \rightarrow X_j^{(T)}$ lies entirely in $L$ except at its endpoints in $X^{(T)}$ and $X^{(t)}$. Concatenating them yields a path $\pi$ from $X_i^{(T)}$ to $X_j^{(T)}$ whose internal vertices are all in $L \cup S^{(<T)}$, and whose only collider in $\tilde{S}$ is (a copy of) $K^{(t)}$, which is an ancestor of the $S^{(t)}$ that is condition on. All other internal vertices are $\epsilon$'s and earlier-generation $X$'s, hence latent. Therefore $\pi$ is an inducing path relative to $(X^{(T)}, S^{(<T)})$, implying adjacency of $X_i^{(T)}$ and $X_j^{(T)}$ in the MAG.

($\Leftarrow$) *If neither $X_i$ and $X_j$ are adjacent in $\mathcal{G}$ nor $\{X_i, X_j\} \subseteq \mathrm{an}_{\mathcal{G}}(S)$, then $X_i$ and $X_j$ are non-adjacent in $\mathcal{E}(\mathcal{G})$.*

Assume $X_i$ and $X_j$ are not adjacent in $\mathcal{G}$ and at least one of them, say $X_j$, is not in $\mathrm{an}_{\mathcal{G}}(S)$. We claim there is no inducing path between $X_i^{(T)}$ and $X_j^{(T)}$ relative to $(X^{(T)}, S^{(<T)})$. Indeed, any path from $X_i^{(T)}$ to $X_j^{(T)}$ that leaves generation $T$ must pass through an $\epsilon$-chain to earlier generations. To create an inducing path that cannot be blocked by variables in $X^{(T)}$, the path must be activated via conditioned colliders in $S^{(<T)}$. But every conditioned selection node $S^{(t)}$ only has ancestors in $X^{(t)}$ that correspond (in the static graph) to $\mathrm{an}_{\mathcal{G}}(S)$. Since $X_j \notin \mathrm{an}_{\mathcal{G}}(S)$, no segment that uses a conditioned $S^{(t)}$ can reach $X_j^{(T)}$ through a directed ancestry pattern required by an inducing path. Consequently any candidate inducing path would have to induce dependence through a direct causal adjacency at time $T$, which is excluded by the assumption that $X_i$ and $X_j$ are non-adjacent in $\mathcal{G}$.

**Step 3: Orientations and absence of undirected edges.**    Given an adjacency $X_i, X_j$ in the MAG, its endpoint marks are directly determined by the latent/selection projection rules (Richardson & Spirtes, 2002). In our setting, two key simplifications occur.

First, *no undirected edges can appear*. In the MAG formalism, undirected edges arise only when an endpoint is an ancestor of a conditioned selection variable. Here, however, the conditioned selection nodes are $S^{(<T)}$, which occur strictly before generation $T$. Because $\mathcal{G}^{(T)}$ is acyclic, no vertex in $X^{(T)}$ can be an ancestor of any $S^{(t)}$ with $t < T$. Hence the "selection ancestor" condition never holds for endpoints in $X^{(T)}$, and therefore $\mathcal{E}(\mathcal{G})$ has no undirected edges.

Second, the remaining endpoint marks coincide with ancestral relations in the static graph $\mathcal{G}$. If $X_i \in \mathrm{an}_{\mathcal{G}}(X_j)$, then the same directed ancestry holds within generation $T$ in $\mathcal{G}^{(T)}$: $X_i^{(T)} \in \mathrm{an}_{\mathcal{G}^{(T)}}(X_j^{(T)})$. By the MAG projection rules, this forbids an arrowhead at $X_i$ on the $i$–$j$ edge and yields $X_i \to X_j$. Symmetrically, if $X_j \in \mathrm{an}_{\mathcal{G}}(X_i)$ we obtain $X_j \to X_i$. If neither is an ancestor of the other in $\mathcal{G}$, the $X_i \leftrightarrow X_j$ edge is also directly given.    $\square$

Then, it is straightforward to go from Theorem A.1 to Theorem 1. Specifically, the clique structure among $\mathrm{an}_{\mathcal{G}}(S)$ induces that no v-structures (in the MAG) exists, and no edges among them can be oriented. Thus just a DAG with same adjacencies in this clique would also suffice the represent all the corresponding d-separations.    $\square$

Regarding this DAG representation, recall that in this work we restrict the number of selection variables as one, for expository clarity: While in static selection settings, it is natural to have multiple selection variables to represent multiple independent selection events, we cannot think of a justification for doing so in the evolutionary setting, where reproduction is the single event to model. This however does not limit generality: the results still extend when $\mathcal{G}$ has multiple selection variables. Specifically, with multi-node $S$, the induced CI relations can still be represented by a DAG with edges augmented among $S$'s ancestors; however, these edges now do not necessarily form a clique, as follows:

**Corollary 2 (Augmented DAG representation for evolutionary models with multiple selection variables).** *Let $\mathcal{G}$ be a DAG over vertices $X \cup S$, and $\pi$ an ordering topological to $\mathcal{G}$. Note that different from the main text, here $S$ may contain multiple variables. The augmented DAG of $\mathcal{G}$, denoted by $\mathcal{G}^+$, is a DAG over $X$, where $X_i \to X_j \in \mathcal{G}^+$ if and only if either $X_i \to X_j \in \mathcal{G}$, or the following three conditions hold: $\{X_i, X_j\} \subseteq \mathrm{an}_{\mathcal{G}}(S)$, $\pi(X_i) < \pi(X_j)$, and $X_i \not\perp\!\!\!\perp_d X_j | S$ in $\mathcal{G}$. Note that the third condition is automatically satisfied when $|S| = 1$. When $|S| > 1$, the augmented edges may not form a clique.*

*Then, $\mathcal{G}^+$ can correctly represent the d-separations implied by $\mathcal{G}^{(T)}$, in a same way as stated in Theorem 1.*

## B.5    Proof of Theorem 2

**Theorem 2 (Soundness and completeness of Algorithm 1).** *Suppose the observed data are large enough i.i.d. samples drawn from $p(X^{(T)}|S^{(<T)} = 1)$, with $X$, $S$, $\mathcal{G}$, $\mathcal{G}^{(T)}$ be as in Definition 1, and assume faithfulness w.r.t. Lemma 2. Let $\mathcal{C}$ be the CPDAG output by Algorithm 1 on data. Then:*

*Soundness and completeness of adjacencies: Two vertices $X_i$ and $X_j$ are adjacent in $\mathcal{C}$, if and only if they are directly causally related, or both involve in selection; i.e., $X_i \in \mathrm{pa}_{\mathcal{G}}(X_j)$, or $X_i \in \mathrm{ch}_{\mathcal{G}}(X_j)$, or $\{X_i, X_j\} \subseteq \mathrm{an}_{\mathcal{G}}(S)$.*

*Soundness of orientations: For any oriented edge $X_i \to X_j \in \mathcal{C}$, this direct causal relation is true, and $X_j$ is not involved in selection, i.e., $X_i \in \mathrm{pa}_{\mathcal{G}}(X_j)$ and $X_j \notin \mathrm{an}_{\mathcal{G}}(S)$.*

*Completeness of orientations: For any unoriented edge $X_i$—$X_j \in \mathcal{C}$, no further identification is possible; i.e., there always exists an alternative DAG $\mathcal{G}'$ that differs from $\mathcal{G}$ in the relation between $X_i, X_j$ (among the three possibilities for adjacency),*

*yet induces the same CPDAG output $\mathcal{C}$.*

*Proof of Theorem 2.* **Soundness of Adjacencies:** Algorithm 1 (PC/GES) is sound and complete for recovering the Markov Equivalence Class (CPDAG) of the distribution's underlying DAG under the faithfulness assumption. By Theorem 1, the CI relations of the data are represented by $\mathcal{G}^+$. Thus, Algorithm 1 outputs the CPDAG of $\mathcal{G}^+$, denoted $\mathcal{C}$. Vertices $X_i, X_j$ are adjacent in $\mathcal{C}$ iff they are adjacent in $\mathcal{G}^+$. By Definition 2, adjacency in $\mathcal{G}^+$ implies either $X_i \to X_j \in \mathcal{G}$ (direct cause) or $\{X_i, X_j\} \subseteq \mathrm{an}_{\mathcal{G}}(S)$ (selection involvement).

**Soundness of Orientations:** In the CPDAG $\mathcal{C}$, an edge $X_i \to X_j$ is oriented if it is part of a v-structure (or propagated from one). Edges within the selection clique $\{X \mid X \in \mathrm{an}_{\mathcal{G}}(S)\}$ form a fully connected subgraph. A fully connected subgraph contains no v-structures (induced subgraphs of form $A \to B \leftarrow C$ with $A, C$ non-adjacent). Therefore, no edges within the selection clique can be oriented by the PC algorithm purely based on internal structure. If $X_i \to X_j$ is oriented in $\mathcal{C}$, it implies $X_j$ cannot be part of the clique (otherwise the edge would be undirected or part of a larger undirected component, unless $X_i$ is outside and forms a v-structure). Specifically, if $X_i \to X_j$ is output, and $X_j$ were in $\mathrm{an}_{\mathcal{G}}(S)$, $X_i$ (as a parent) would also be in $\mathrm{an}_{\mathcal{G}}(S)$, placing the edge inside the clique, rendering it unoriented. Thus, oriented edges imply true causal relations where the effect is not involved in selection.

**Completeness of Orientations:** As noted above, the selection-induced edges form a clique. In a Markov Equivalence Class, edges within a fully connected clique cannot be distinguished from one another without external v-structures. Since the added edges in $\mathcal{G}^+$ mimic causal edges but form a clique, they are structurally indistinguishable from a dense set of causal relations. Thus, they remain unoriented in $\mathcal{C}$. Specifically note that selection existence cannot be confirmed, since this augmented DAG itself contains no selection and suffices to represent all the d-separations. □

### B.6   Proof of Theorem 3 and Theorem 4

**Theorem 3** (**Multi-domain clique-augmented DAG representation**). *Let all notations be as in Definitions 1 to 3. We construct a new DAG $\mathcal{G}^{+I}$ by adding an auxiliary vertex $\zeta$ to $\mathcal{G}^+$ and draw an edge $\zeta \to X_i$ for each $X_i \in I$. Further, if $\mathrm{an}_{\mathcal{G}}(S) \cap I \neq \varnothing$, that is, when selection itself or any variable involved in selection is changed, we add edges from $\zeta$ to all of $\mathrm{an}_{\mathcal{G}}(S) \backslash \{S\}$ to $\mathcal{G}^{+I}$. Then, for any sets $A, C \subseteq X$, a d-separation $\zeta \perp\!\!\!\perp_d X_A \mid X_C$ in $\mathcal{G}^{+I}$ implies invariance of the observed conditional distribution $p^{(k)}(X_A^{(T_k)} \mid X_C^{(T_k)}, S^{(<T_k)} = 1)$ across all the $K$ domains.*

**Theorem 4** (**Soundness and completeness of Algorithm 2**). *Let all notations be as in Definitions 1 to 3. Assume faithfulness w.r.t. Lemma 2 and Theorem 3. Let $\mathcal{P}_X$ be the subgraph of Algorithm 2's output PDAG on $X$. Then, all statements of Theorem 2 continue to hold when '$\mathcal{C}$' is replaced by '$\mathcal{P}_X$'; $X_i \to X_j \in \mathcal{C}$ implies $X_i \to X_j \in \mathcal{P}_X$, but not vice versa.*

Proof of Theorem 3 and Theorem 4 follows directly from those of Theorem 1 and Theorem 2, by introducing an exogenous domain index variable (the "change driver") $\zeta$ into the evolutionary graph. Then, the invariance of a conditional distribution $p(X_A \mid X_C)$ can be expressed as the CI relation $\zeta \perp\!\!\!\perp X_A \mid X_C$, which can further be read off from the corresponding d-separation in the evolutionary graph. Finally, we characterize these d-separations via a DAG, leading to Theorem 3. For Theorem 4, it is because that PC, or more specifically, Meek rules R4 (which is used in CD-NOD; (Huang et al., 2020)) is proved to be complete under such prior knowledge (all edges adjacent to $\zeta$ must be outgoing), the soundness and completeness all follow.

## C   Related Work

Research on selection bias broadly follows two complementary directions: (i) *nonparametric* approaches that exploit conditional independence constraints implied by selection mechanisms, and (ii) *parametric* approaches that explicitly model the selection process within a causal framework.

**Causal discovery under selection.**   Building on the FCI algorithm, a line of nonparametric methods studies how causal structure can be recovered in the presence of selection bias using conditional independence information (Hernán et al., 2004; Tillman & Spirtes, 2011; Evans & Didelez, 2015; Akbari et al., 2021; Versteeg et al., 2022). More recent work has examined selection bias in interventional settings (Dai et al., 2025a; Luo et al., 2026), in latent-variable settings (Dai et al., 2025b; Luo et al., 2025), as well as in temporal and sequential data (Zheng et al., 2024; Qiu et al., 2024). Parametric approaches have also been proposed, particularly for bivariate causal orientation under selection (Zhang et al., 2016; Kaltenpoth & Vreeken,

2023). Selection mechanisms has also been used to guide text-generation tasks (Tang et al., 2025; Deng et al., 2026).

**Causal inference and bias adjustment.** From an inferential perspective, nonparametric work has centered on the selection diagram framework, which provides graphical conditions under which causal effects remain identifiable despite selection bias (Bareinboim & Pearl, 2012; Bareinboim et al., 2014; Bareinboim & Tian, 2015; Bareinboim & Pearl, 2016; Bareinboim et al., 2022). Subsequent studies derived testable implications of selection mechanisms and corresponding adjustment criteria (Correa et al., 2019). In parallel, parametric treatments of selection bias have a long history in economics and biostatistics, most notably through selection models and related corrections (Heckman, 1977; Dubin & Rivers, 1989; Heckman, 1990; Winship & Mare, 1992; Robins et al., 2000).

**Connections to missing data.** Selection bias is closely related to problems of data missingness. In particular, under certain missingness mechanisms such as self-masking, both the full data distribution and causal effects may be unidentifiable, making causal inference infeasible (Mohan et al., 2013; Mohan, 2018; Tu et al., 2019; Miao et al., 2016; Gao et al., 2022; Enders, 2022). Nonetheless, in such settings, it is sometimes still possible to recover aspects of the underlying causal structure (Tu et al., 2019; Dai et al., 2024).

**Evolution and causal modeling.** A related literature is called evolutionary or reciprocal causation (Oyama et al., 2003; Uller & Lala, 2019), but it focuses primarily on philosophical aspects rather than formal graphical treatments. Graphical models and structural equation models are used in (Edelaar et al., 2023; Baedke, 2021; Ramsey, 2016), but not for the purpose of causal discovery. Several works also study evolutionary game theory and its causal interpretation (Lehtonen & Otsuka, 2023).

# D Supplementary Experimental Details and Results

## D.1 Additional Experimental Results on Synthetic Data

We now elaborate more on the simulation experiments, in particular about the reproduction step. For each generation $t$, we rank samples according to their $S$ value. Their ranks are then categorized into 6 uniform segments, corresponding to having $0, 1, \ldots, 5$ offspring. This would result the next generation to have a $\sim 2.5\times$ sample size than the current generation. We thus then do a random downsampling on this next generation so that the sample size is fixed across generations. In our experiments, we use the sample size $N = 5,000$.

We use the PC and GES implementations from causal-learn (Zheng et al., 2023), with the alpha set to $0.05$ and L0 penalty set to 2. The precision is calculated based on the true causal adjacencies, regardless of directions.

In light of the suggestion from anonymous reviewers, in addition to PC and GES, we also compare with RLCD (Dong et al., 2024), a method for latent-variable causal discovery (since our evolutionary graph involves latent variables). We also present results under model misspecifications with i) latent confounding, ii) dependent inheritance, and iii) structural changes across domains (edge reversals). We find that: 1) Though designed to model latent variables, RLCD still fails to recover the true causal relations under evolutionary selection, and 2) While misspecifications degrade performance overall, our interpretation still outperforms the standard one, except in extreme cases where result graphs become overly dense.

Detailed results are shown below:

### D.1.1 Comparison with Latent-Variable Causal Discovery Methods

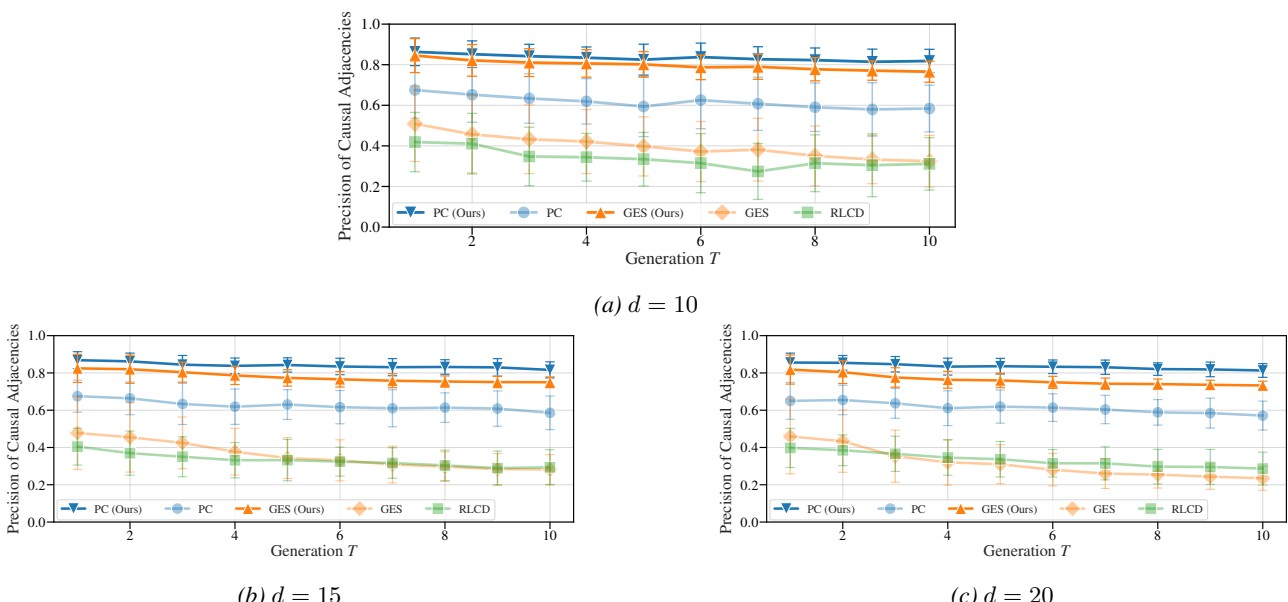

*(a) $d = 10$*

*(b) $d = 15$*  *(c) $d = 20$*

*Figure 9.* Precision (y-axis) of causal adjacencies for $d = 10, 15, 20$ variables, using the same data simulation setup as in Figure 6. We compare the standard interpretation and ours for PC and GES, and additionally include RLCD (Dong et al., 2024), a latent-variable causal discovery method. Mean and standard deviation over 50 random runs are shown. According to the results, though designed to model latent variables, RLCD still fails to recover the true causal relations among observed variables under evolutionary selection.

### D.1.2 Results under Model Misspecification with Violation of Causal Sufficiency

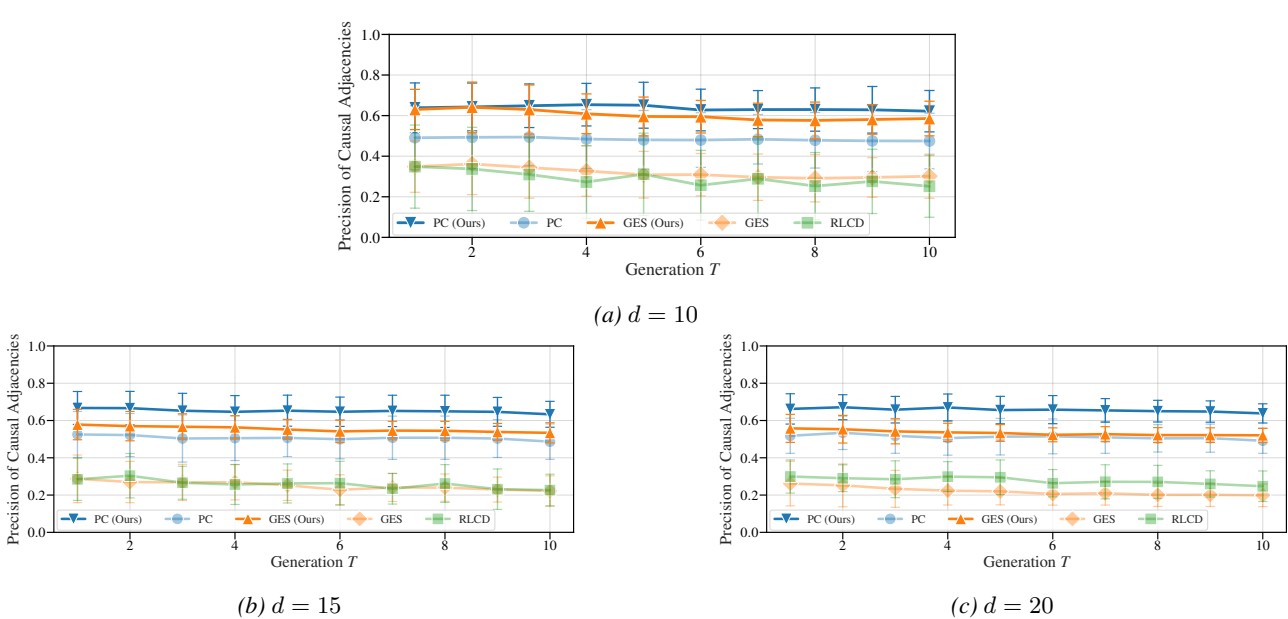

*(a) $d = 10$*

*(b) $d = 15$*  *(c) $d = 20$*

*Figure 10.* Precision (y-axis) of causal adjacencies under a similar setup as in Figure 6. The difference is that, to simulate hidden confounding, we hold out the top $d/5$ variables in the topological ordering of the truth DAG $\mathcal{G}$, and run PC/GES/RLCD on the data of the remaining variables. According to the results, although the overall adjacency precision decreases due to additional spurious dependencies from hidden confounding, our interpretation still improves over the standard one, i.e., oriented edges are "more trustable" than unoriented edges in the CPDAG output.

### D.1.3 Results under Model Misspecification with Dependent Heritable Factors and Non-Component-Wise Effects

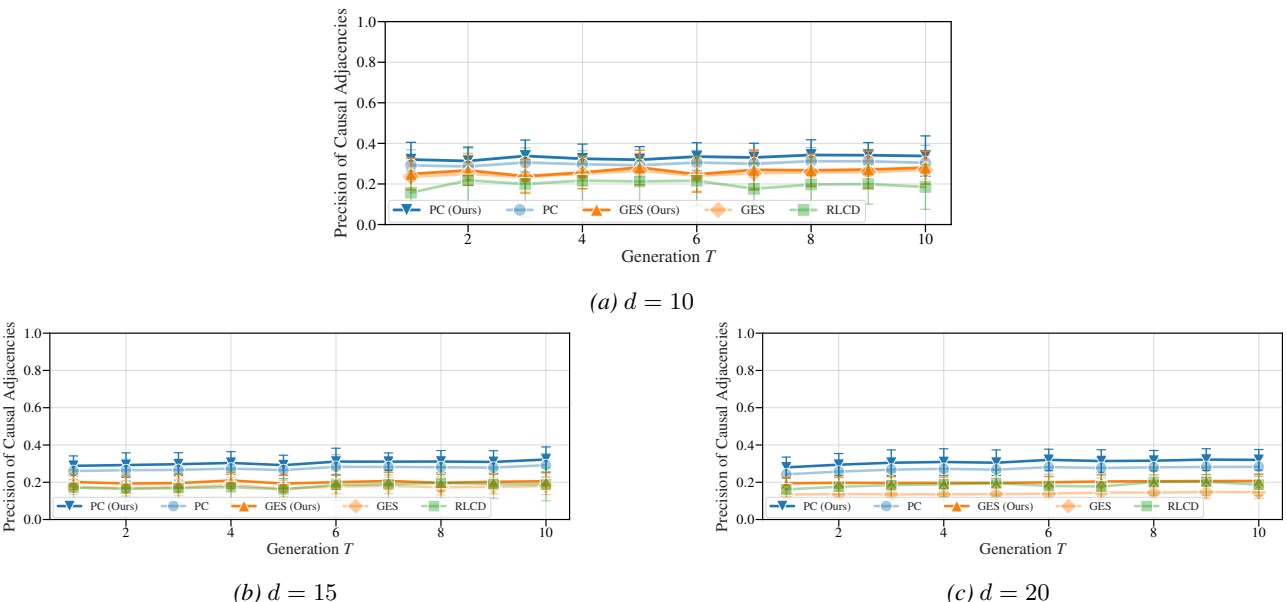

*Figure 11.* Precision (y-axis) of causal adjacencies under a similar setup as in Figure 6. The difference is that, we now allow additional causal relations among the heritable factors $\epsilon^{(t)}$ (e.g., gene regulatory relations), and non-component-wise causal relations from $\epsilon^{(t)}$ to $X^{(t)}$ (e.g., multiple genes jointly influencing a phenotype). The average in-degree of these edges is 2. The result shows that none of these methods or interpretations can adequately handle this setting. This is expected: the CI relations in observed data can no longer be represented by a DAG, and the result graphs now become overly dense due to the lack of informative CI relations left in observed data.

### D.1.4 Results under Model Misspecification with Structural Changes (Edge Reversals) Across Domains

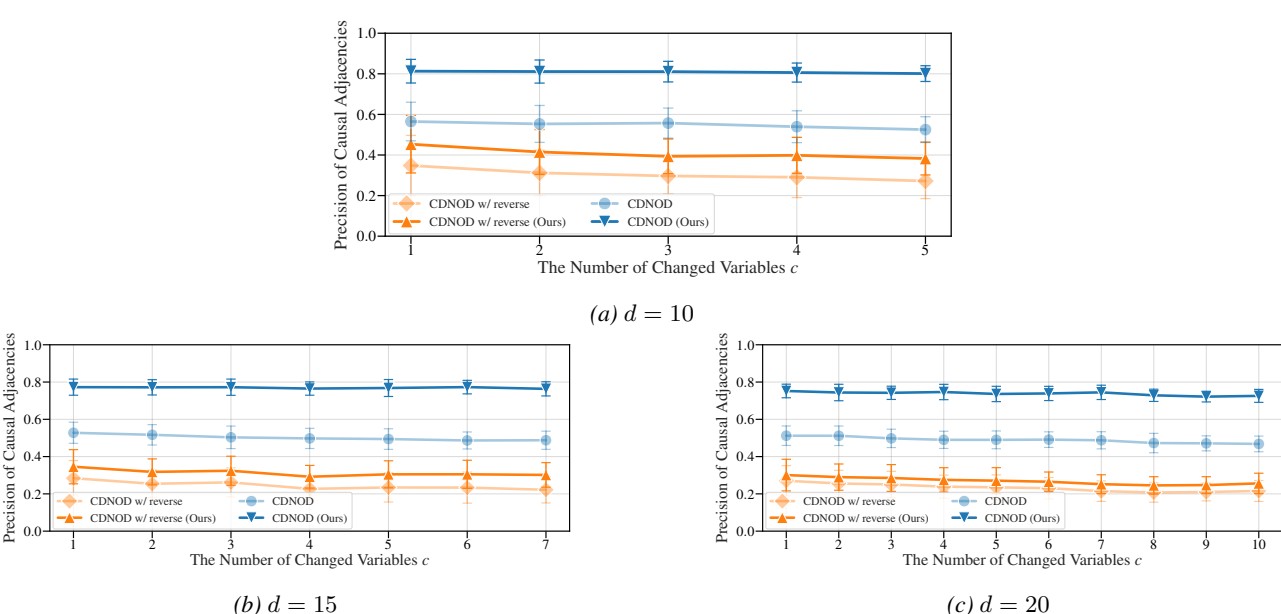

*Figure 12.* Precision (y-axis) of causal adjacencies under the standard interpretation (using all PDAG edges) and our interpretation (using only oriented PDAG edges) of CDNOD outpus on multi-domain data. For each value of $c$ (x-axis), we randomly sample $c$ variables from $X \cup \{S\}$ as changing variables. For each changing variable, we perturb its exogenous noise variance and the linear weights from all its parents. Further, under the model misspecification setting ("w/ reverse"), we additionally reverse the direction of edges by randomly turning half of the changing variable's parents into its children. According to the results, although the overall adjacency precision decreases due to structural changes across domains (in particular the edge reversals), our interpretation still improves over the standard one.

## D.2 Additional Experimental Results on Real-World Data

We then provide more details and the full results on the real-world datasets.

### D.2.1 Results on DGRP, a Gene Expression Dataset with Partially Known Ground Truth

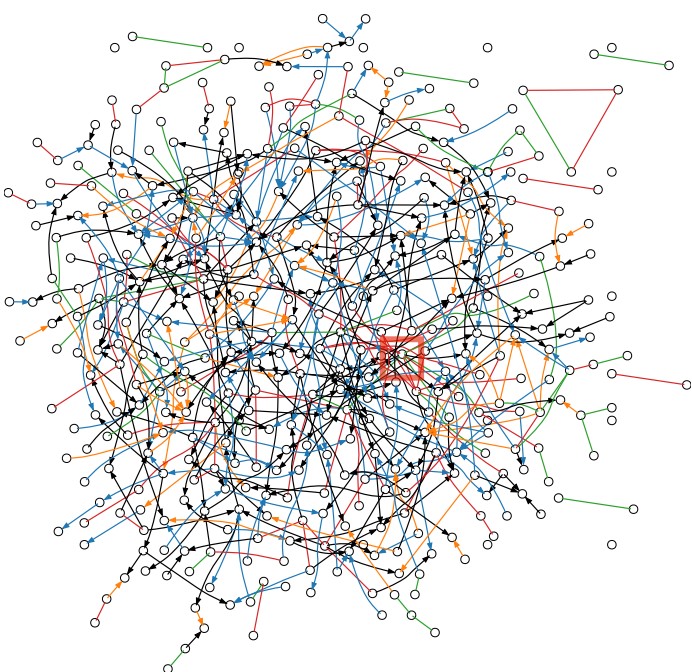

| Edge type | Precision on adjacencies |
|---|---|
| **Oriented** | $49.0\% = (165 + 65)/(165 + 65 + 239)$ |
| **Unoriented** | $35.5\% = 49/(49 + 89)$ |

*Figure 13.* Experiments on the female *Drosophila melanogaster* Genetic Reference Panel (**DGRP**) dataset (Everett et al., 2020). The data consist of bulk RNA-seq measurements on 422 genes for 400 inbred lines. As partially known regulatory ground truth, we use the reported 408 regulator-target pairs annotated by cis-trans expression quantitative trait loci (eQTL) analysis (Everett et al., 2020). **Left:** The CPDAG output by BOSS (Andrews et al., 2023). Oriented edges are colored as follows: blue ($\rightarrow$) for known oriented edges; orange ($\rightarrow$) for edges with known adjacency but reversed orientation; and black ($\rightarrow$) for edges without known adjacency. Unoriented edges are colored as follows: green (—) for known adjacencies, and red (—) for unknown adjacencies. The node highlighted by the red box corresponds to *FBgn0033978*. It appears in multiple unoriented cliques, and this gene has been reported to be related to fitness by involving in detoxification and insecticide resistance (Battlay et al., 2018; Duneau et al., 2018). **Right:** The precision of recovered adjacencies w.r.t. the ground truth. According to the result, oriented edges have a higher hit rate on known regulatory adjacencies (49.0%) than unoriented edges (35.5%). Although eQTL-derived ground truth may be incomplete, this result can, to some extent, support our conclusion that oriented edges are "more trustable" than unoriented ones in a CPDAG output, when evolutionary selection presents.

### D.2.2 Full Results on Remaining Six Datasets with LLM-Annotated Pseudo Ground Truth

For the remaining six datasets that we were unable to find ground-truth graphs, we provide variable names to ChatGPT and ask it to search the literature to annotate i) direct causal relations, and ii) variables directly involved in fitness selection. Using these annotations as pseudo ground truth, we evaluate:

1. Whether oriented edges in results are truly causal: We find that on 5 out of 6 datasets, oriented edges indeed have higher precision on causal adjacencies than unoriented edges.

2. Whether variables involved in selection (i.e., ancestors of $S$) truly form an unoriented clique: However, this is not as expected; the recall of such unoriented edges is consistently low (<= 50%), and we do not yet have a clear explanation.

Of course, we are aware that the reliability of this procedure is not guaranteed, and we treat it only as a surrogate.

Dataset descriptions and full results, including algorithm output, our interpretation, and evaluation against LLM-generated ground truths, are provided below.

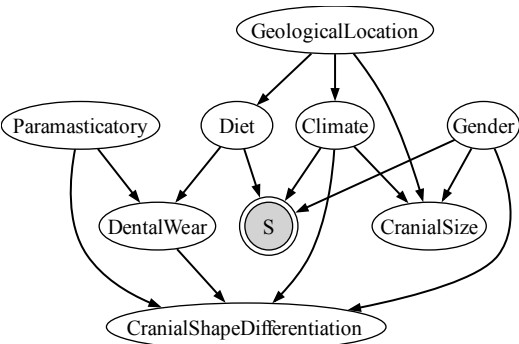

*(a)* ChatGPT annotated pseudo ground truth.

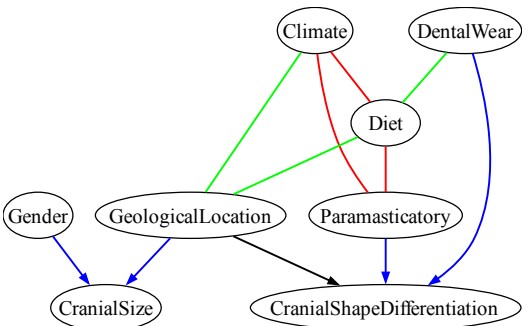

*(b)* PC CPDAG output, with edge colors interpreted in the same way as in Figure 13.

| Edge type | Precision on adjacencies |
|:---:|:---:|
| **Oriented** | $80.0\% = (4 + 0)/(4 + 0 + 1)$ |
| **Unoriented** | $50.0\% = 3/(3 + 3)$ |

*(c)* Precision on the recovered adjacencies.

**Recall on S ancestors**

$50.0\% = 3/(4 \times 3/2)$

*(d)* Recall on the S ancestors. Denominator is the number of unoriented edges that should appear within the clique of S ancestors. Nominator is the number of these unoriented edges that indeed appear in the output.

*Figure 14.* Result on the **Cranial** dataset. Detailed description of this dataset is provided in Appendix D of the original submission. We process this dataset as in (Huang et al., 2018). This dataset contains 8 variables: Gender (1 dimension, discrete), Cranial size (1 dimension, continuous), Diet (5 dimensions, discrete), Paramasticatory behavior (1 dimension, discrete), Dental wear (2 dimensions, mixed continuous and discrete), Geographic location per population (3 dimensions, discrete), Climate per population (6 dimensions, discrete), and Cranial shape differentiation (4 dimensions, continuous). It has a sample size of 255. We use PC with Kernel-based conditional independence test (Zhang et al., 2012) to obtain the result.

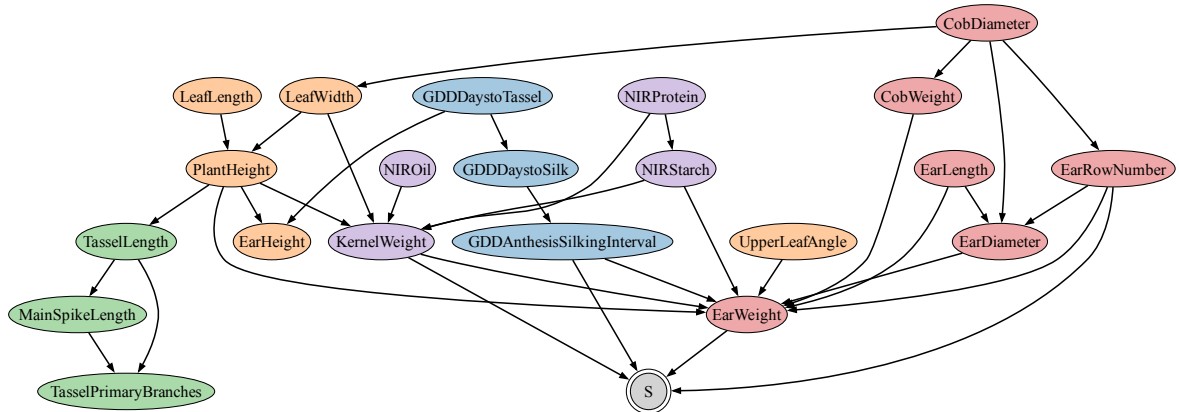

*(a)* ChatGPT annotated pseudo ground truth.

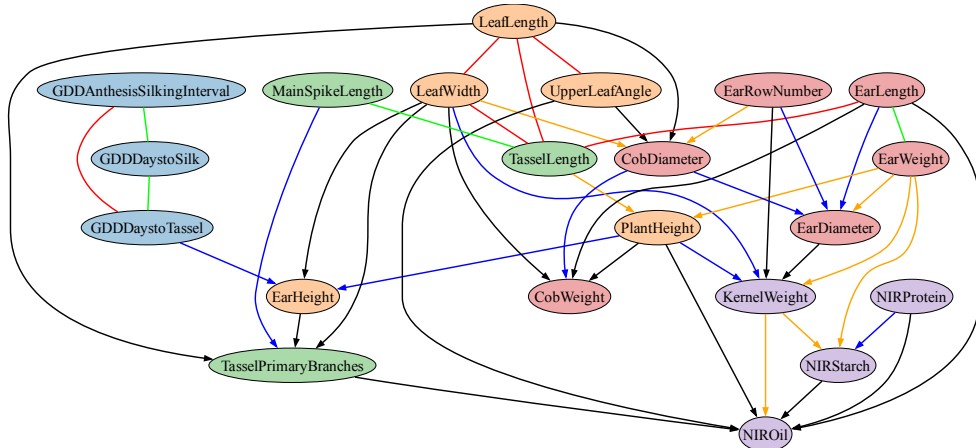

*(b)* CDNOD PDAG output with domain index omitted. Edge colors interpreted as in Figure 13.

| Edge type | Precision on adjacencies |
|---|---|
| **Oriented** | $52.8\% = (10 + 9)/(10 + 9 + 17)$ |
| **Unoriented** | $40.0\% = 4/(4 + 6)$ |

*(c)* Precision on the recovered adjacencies.

**Recall on S ancestors**

$4.4\% = 6/(17 \times 16/2)$

*(d)* Recall on the S ancestors. Denominator is the number of unoriented edges that should appear within the clique of S ancestors. Nominator is the number of these unoriented edges that indeed appear in the output.

*Figure 15.* Result on the **Panzea** dataset, where "PlantEnvironment" is treated as the domain index. Different colors of nodes correspond to their different categories: Phenology / Developmental Timing, Plant Architecture (Vegetative Structure), Tassel (Male Inflorescence) Traits, Ear & Cob Morphology (Female Inflorescence Structure), and Kernel Composition (NIR-based Quality Traits). We use 4 different environments: maize planted in Aurora, NY in summers 2006 and 2007, in Clayton, NC in summer 2006, and in Ponce, PR in winter 2006. In total there are 22 variables (including the plant environment), and 9,326 samples. Since these are the numerical/ordinal variables, we use PC with Fisher's Z-test.

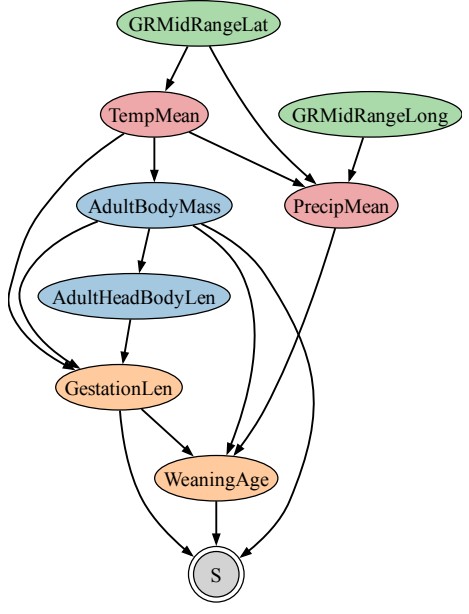

*(a)* ChatGPT annotated pseudo ground truth.

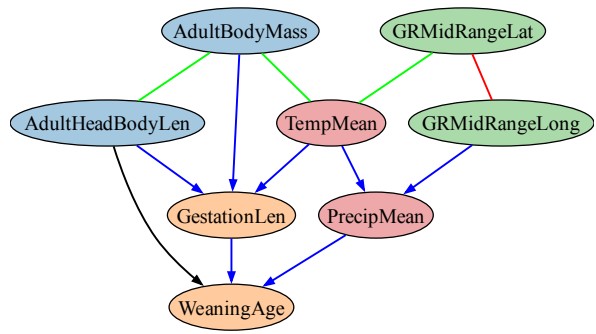

*(b)* PC CPDAG output, with edge colors interpreted in the same way as in Figure 13.

| Edge type | Precision on adjacencies |
|-----------|--------------------------|
| **Oriented** | $87.5\% = (7 + 0)/(7 + 0 + 1)$ |
| **Unoriented** | $75.0\% = 3/(3 + 1)$ |

*(c)* Precision on the recovered adjacencies.

**Recall on S ancestors**

$14.3\% = 4/(8 \times 7/2)$

*(d)* Recall on the S ancestors. Denominator is the number of unoriented edges that should appear within the clique of S ancestors. Nominator is the number of these unoriented edges that indeed appear in the output.

*Figure 16.* Result on the **PanTHERIA** dataset. Different colors of nodes correspond to their different categories: Body Size Morphology, Life History Timing, Geographic Range, and Climate Environment. We use 8 variables that will not lead to too many missingness. The sample size used is 626. Since these are the numerical/ordinal variables, we use PC with Fisher's Z-test.

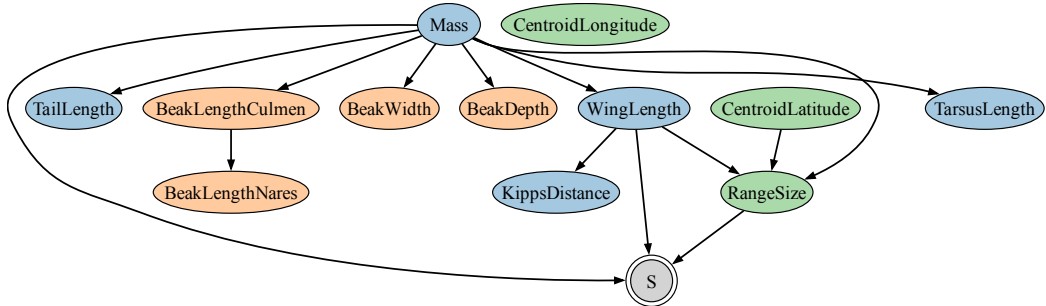

*(a)* ChatGPT annotated pseudo ground truth.

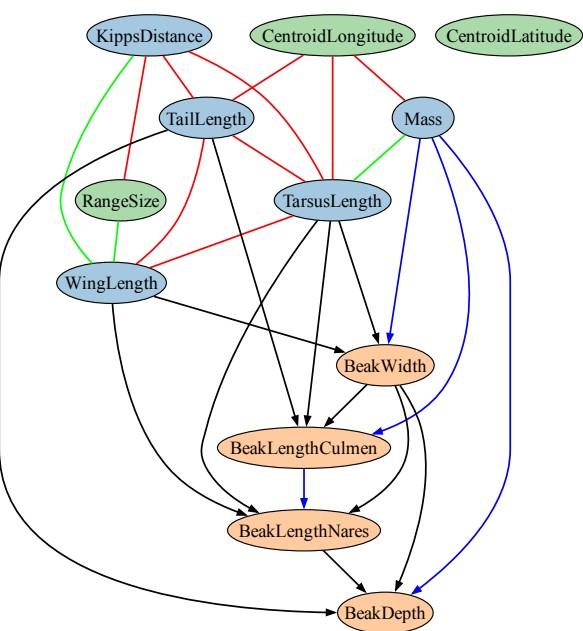

*(b)* PC CPDAG output, with edge colors interpreted in the same way as in Figure 13.

| Edge type | Precision on adjacencies |
|-----------|--------------------------|
| **Oriented** | $26.7\% = (4 + 0)/(4 + 0 + 11)$ |
| **Unoriented** | $25.0\% = 3/(3 + 9)$ |

*(c)* Precision on the recovered adjacencies.

**Recall on S ancestors**

$16.7\% = 1/(4 \times 3/2)$

*(d)* Recall on the S ancestors. Denominator is the number of unoriented edges that should appear within the clique of S ancestors. Nominator is the number of these unoriented edges that indeed appear in the output.

*Figure 17.* Result on the **AVONET** dataset. Different colors of nodes correspond to their different categories: Body Size Morphology, BeakSize Morphology, and Geographic Range. There are 12 variables and 10,950 samples. Since these are the numerical/ordinal variables, we use PC with Fisher's Z-test. Other variables, like diet, may be useful and informative when included. But we leave them because they are categorical coded.

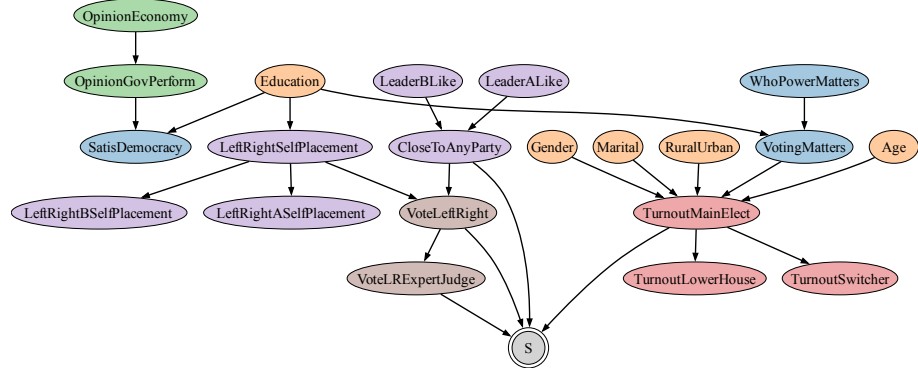

*(a)* ChatGPT annotated pseudo ground truth.

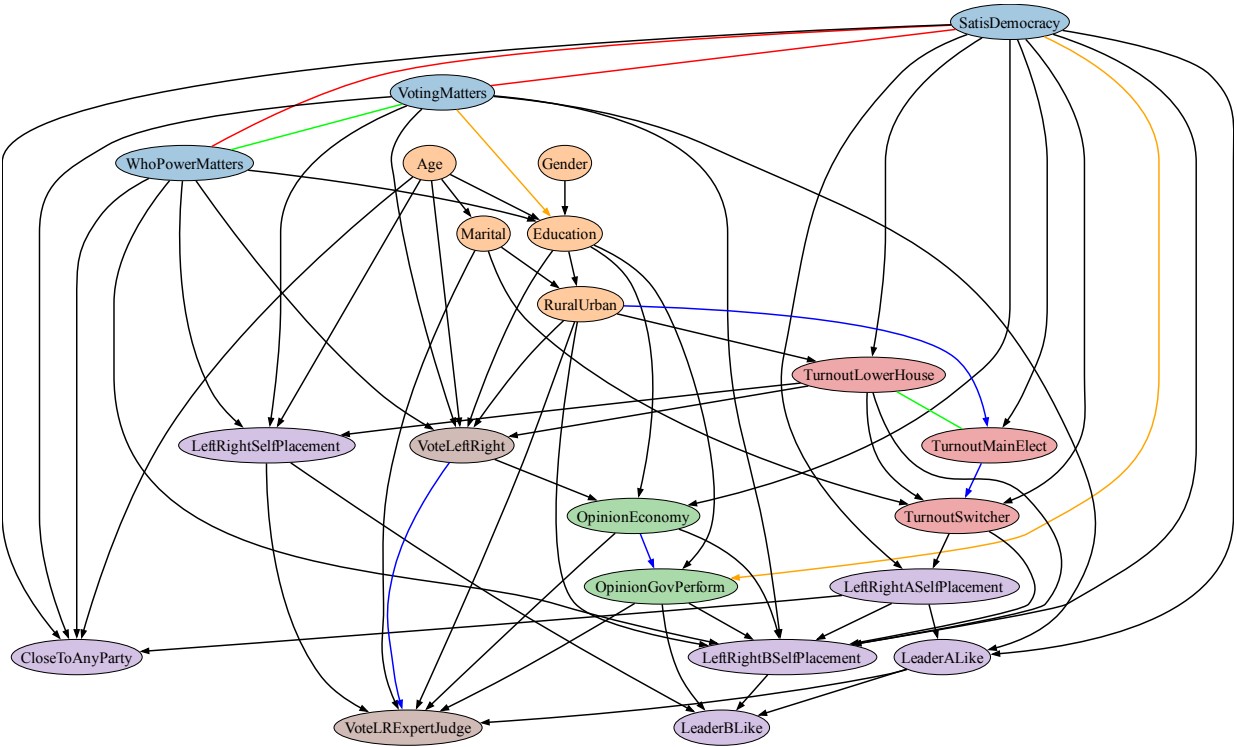

*(b)* CDNOD PDAG output with domain index omitted. Edge colors interpreted as in Figure 13.

| Edge type | Precision on adjacencies |
|-----------|--------------------------|
| **Oriented** | $9.8\% = (4 + 2)/(4 + 2 + 55)$ |
| **Unoriented** | $50.0\% = 2/(2 + 2)$ |

*(c)* Precision on the recovered adjacencies.

**Recall on S ancestors**

$1.1\% = 1/(14 \times 13/2)$

*(d)* Recall on the S ancestors. Denominator is the number of unoriented edges that should appear within the clique of S ancestors. Nominator is the number of these unoriented edges that indeed appear in the output.

*Figure 18.* Result on the **CSES** dataset, where "Country" is treated as the domain index. Different colors of nodes correspond to their different categories: Democracy, Demographics, Economy, Vote Engagement (Turnour Behavior), Ideology, Left/Right Self Placements, and Vote Ideology. We use data from 5 countries: Germany, India, Switzerland, Netherlands, and United States of America. In total there are 22 variables (including country), and 15,269 samples. We use PC with Fisher's Z-test.

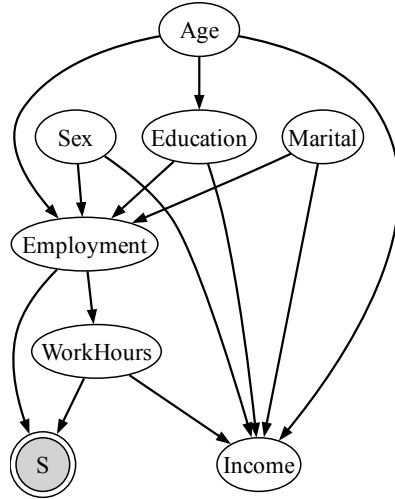

*(a)* ChatGPT annotated pseudo ground truth.

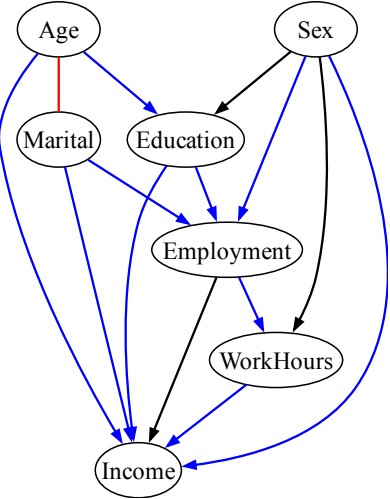

*(b)* PC CPDAG output, with edge colors interpreted in the same way as in Figure 13.

| Edge type | Precision on adjacencies |
|:---:|:---:|
| **Oriented** | $76.9\% = (10 + 0)/(10 + 0 + 3)$ |
| **Unoriented** | $0.0\% = 0/(0 + 1)$ |

*(c)* Precision on the recovered adjacencies.

| Recall on S ancestors |
|:---:|
| $6.7\% = 1/(6 \times 5/2)$ |

*(d)* Recall on the S ancestors. Denominator is the number of unoriented edges that should appear within the clique of S ancestors. Nominator is the number of these unoriented edges that indeed appear in the output.

*Figure 19.* Result on the **PUMS** dataset. Most of the available variables are categorical; we curate 7 of them that are continuous/ordinal. We use the data collected in 2024 in 50 states. The total sample size is 367,858. We use PC with Fisher's Z-test. In this result, all but one edges are oriented, suggesting the less likelihood of selection in evolution (under our model class).

