# OpenReview forum: "Causal Modeling of Selection in Evolution"
_ICML.cc/2026/Conference — ICML 2026 spotlight_

### Official Review · Reviewer_4yqK · 2026-03-09

**Soundness:** 3
**Presentation:** 3
**Significance:** 3
**Originality:** 4
**Overall Recommendation:** 5
**Confidence:** 3

**Summary:**

This paper distinguishes between static selection, where observed data arise from a one-shot filtering process, and evolutionary selection, where observed data correspond to a generation shaped by repeated selection and reproduction. The main claim is that the standard static selection graph is appropriate for the former but can be misleading for the latter. To address this, the paper introduces an evolutionary selection model, derives its Markov properties, and presents sound and complete identification procedures for both single-domain and multi-domain data. A notable aspect is that estimation reduces to running standard causal discovery methods such as PC/GES or CDNOD, but with a different interpretation of the output under evolutionary selection. The paper evaluates the approach on synthetic data and several biological and social datasets.

**Compliance With Llm Reviewing Policy:**

Affirmed.

**Final Justification:**

The rebuttal adequately addressed my concerns, and I am updating my recommendation to Accept. The additional analyses helped to strengthen the paper.

**Key Questions For Authors:**

1.  In the multi-domain setting, the graph structure is assumed fixed while only mechanism parameterizations change across domains. How restrictive is this assumption in the intended application areas, and is there a natural extension to structural changes across domains?

2. A central result is that evolutionary selection cannot be confirmed from observational CI information alone. In practice, what signals should guide a user to prefer the evolutionary-selection interpretation over a standard causal or static-selection one?

3. The framework assumes causal sufficiency with respect to the observed traits and also assumes independent heritable factors. How sensitive are the identification results and resulting interpretations to violations of these assumptions, such as interacting genes or unobserved common causes among observed traits?

**Limitations:**

No. The paper does acknowledge one important limitation, namely the simplified modeling of inheritance. However, the limitations discussion could be more explicit about the reliance on causal sufficiency, the fixed-structure assumption in the multi-domain setting, and the fact that the real-data validation is primarily qualitative.

**Strengths And Weaknesses:**

This paper addresses an interesting and underexplored problem. The distinction between static and evolutionary selection is conceptually useful, and the paper makes a credible case that conflating the two can lead to incorrect causal interpretations. The technical development is substantial. The evolutionary selection model is clearly defined, the clique-augmented DAG representation is elegant, and the identification results for both single-domain and multi-domain settings are a real strength. I also like that the resulting procedure is operationally simple. One can apply standard causal discovery tools, but the paper carefully explains why their outputs must be interpreted differently in this setting. The empirical section is also reasonably broad. The synthetic experiment is closely aligned with the theory and supports the main claim that a more conservative interpretation of CPDAG outputs is preferable to the standard one under evolutionary selection. The real-data section spans both biological and social datasets, and the learned structures are at least qualitatively interpretable. My main reservations are about the practical scope of the framework. The theory still relies on standard assumptions such as causal sufficiency, and the model makes a simplifying independence assumption on the heritable factors. I understand why this latter assumption is made for identifiability, but the current biological justification is overstated. In most biological systems, heritable factors are strongly interdependent, so the paper should present this more clearly as a simplification rather than as a realistic biological premise. In addition, the multi-domain results assume that the causal and selection graph structures remain fixed and only parameterizations change across domains; this may be restrictive for some real evolutionary settings. On the empirical side, the real-world data results are interesting but mostly qualitative, so the evidence for practical usefulness is suggestive rather than decisive. Finally, an important point in the theory is that evolutionary selection can be falsified but not confirmed from observational CI information alone. That is a valuable result, but it also means that clearer guidance on when practitioners should adopt the evolutionary-selection interpretation would strengthen the paper.

---

> ### Author Rebuttal · Authors · 2026-03-31
>
> We are grateful for the reviewer's helpful feedback. Please see below for our response.
>
> ---
> **(Q1)** The reviewer wonders whether there is a natural extension to structural changes across domains.
>
> **A:** Thank you for this insightful question. The answer is yes.
>
> To see this, we first note that our current results can already support a specific kind of structural change, namely _edge vanishing_: An edge $X\to Y$ appearing in one domain but not in another can also be viewed as a parameter change (from nonzero to zero) of this edge, i.e., a mechanism change for $Y$. In other words, as long as the union graph across domains remains acyclic, our current results still apply.
>
> Then, to handle more general structural changes with _edge reversals_ ($X\to Y$ in one domain and $Y\to X$ in another), we can similarly view the underlying graph as containing both edges (a 2-loop), where one edge vanishes in each domain, i.e., mechanism changes for both $X$ and $Y$. This reduces the problem to cyclic causal discovery. We avoid this setting for simplicity, as cyclic models are more complex to learn and interpret. However, their theoretical results do already exist [[R96]](https://tinyurl.com/mwyvf2vc)[[MC20]](https://tinyurl.com/32e85ums).
>
> We have expanded the discussion on this in the limitations section in the revision, and have also added synthetic experiments under such model misspecification; see our response to Q4.
>
> ---
> **(Q2)** The reviewer wonders what practical signals would support the evolutionary-selection interpretation, given that it cannot be theoretically confirmed.
>
> **A:** Thank you for this question. Indeed, since there always exists an alternative DAG without selection that induces the same set of CI relations in the data (Theorem 2), by using CI relations alone, evolutionary selection can only be falsified (i.e., if there is selection, its ancestors must lie within a restricted set), but not confirmed (i.e., one cannot conclude that there must be selection).
>
> That said, in practice, when an unoriented clique (or "almost" clique) appears in the CPDAG output in which pairwise direct causal relations are implausible (e.g., based on domain knowledge), and especially when this clique is large and located upstream, these variables' joint involvement in evolutionary selection can be a compelling alternative explanation. The clique _{KippsDistance, TailLength, WingLength, TarsusLength}_ in the _AVONET_ result may be such an example.
>
> ---
> **(Q3)** The reviewer asks about our model's sensitivity to violations of causal sufficiency and the _independent inheritance_ assumption.
>
> **A:** Thanks for this important point. Below we clarify which parts of our theoretical results extend, which do not, and what this implies for possible generalizations. We have also added synthetic experiments under such violations; see Q4.
>
> When heritable factors $\epsilon$ are dependent (e.g., epistasis) and their effects on traits $X$ are non-component-wise (e.g., pleiotropy), or when $X$ is not fully observed, our graphical model remains valid. In particular, Theorem 1 and Lemma 1 still hold: The evolutionary graph (with additional $\epsilon$-edges) still represents the CI relations in $X$, and evolution still induces only additional conditional dependencies, not independencies.
>
> However, these additional dependencies can no longer be represented by a clique-augmented DAG, so Theorem 2 and subsequent identification results no longer apply. While they can still be represented via MAG/FCI, as suggested by additional experiments, such representations may be less informative in practice, due to fewer usable CI relations.
>
> Therefore, to improve identifiability, stronger and possibly equilibrium-aware parametric assumptions may be needed to exploit information beyond CI, as those already used in latent-variable causal discovery and latent state-space models. We hope that our model can serve as a useful starting point in such potential explorations.
>
> ---
> **(Q4)** The reviewer has concerns that the real-data validation is primarily qualitative.
>
> **A:** Thank you for this great point. We actually share the same concern, and put a lot of efforts trying to find evolutionary-selection-related datasets with ground-truth causal graphs for evaluation. Unfortunately, we were unable to find any.
>
> That said, to make the evaluation more convincing, we have added the following analyses:
> - Results on a new dataset (DGRP RNA-seq) with partial ground-truth regulatory relations;
> - Evaluation with LLM-generated pseudo ground truth on the original six real datasets; and
> - Extended simulations with additional baselines and under model misspecifications (latent confounding, dependent inheritance, and structural change).
>
> Detailed results are available at **[[this anonymous link]](https://tinyurl.com/msyfxa8v)**. For more discussion, please kindly refer to our response to Reviewer yMH2's Q1.
>
> ---
> We want to thank the reviewer again for the insightful feedback.

---

> > ### Author Rebuttal · Reviewer_4yqK · 2026-04-02
> >
> > Each of my concerns has been systematically addressed by the authors and I would recommend this paper for acceptance.

---

### Official Review · Reviewer_AMZw · 2026-03-13

**Soundness:** 3
**Presentation:** 3
**Significance:** 3
**Originality:** 4
**Overall Recommendation:** 4
**Confidence:** 3

**Summary:**

This paper studies causal discovery under selection and argues that it is important to distinguish between static selection and evolutionary selection. The main claim is that standard static selection models may be appropriate for one-shot filtering settings, but may not correctly capture the dependencies induced by repeated reproduction and selection across generations. To address this, the authors introduce an evolutionary selection model, analyze its conditional independence structure, and propose a graphical representation that allows the outputs of standard causal discovery methods to be reinterpreted in this setting. The paper also extends the discussion to multi-domain data and provides synthetic and real-data experiments.

**Compliance With Llm Reviewing Policy:**

Affirmed.

**Key Questions For Authors:**

1. In the real-data section, some conclusions are phrased fairly strongly. For example, the PUMS result is said to suggest absence of selection in evolution because all edges are oriented. Is that intended as a strong substantive claim, or only as an implication within the model class assumed by the paper? Clarifying this would make the empirical interpretation more precise.

2. The paper mentions that the inheritance model is simplified. Could the authors say a bit more concretely which part of the main result depends most on this assumption? For example, is it mainly needed for the clique-augmented interpretation, or already for the earlier CI characterization?

3. Since the synthetic study mainly compares your interpretation with the standard reading of PC/GES, do you think it would be useful to also show what happens if one applies a more latent-variable-oriented pipeline on the same data, even just as a sanity check?

**Limitations:**

Yes. The paper does acknowledge important modeling limitations, especially regarding the simplified inheritance structure. I appreciated that this was stated explicitly rather than hidden in the appendix.

**Strengths And Weaknesses:**

Strengths

The paper has a clear and interesting main idea. The distinction between static selection and evolutionary selection is not just terminological; the manuscript gives a concrete example showing that a CI relation read off from the static graph can fail once one accounts for inheritance across generations. That part made the motivation much more convincing to me than if the paper had only argued at an intuitive level. The move from the full evolutionary graph to the clique-augmented DAG is also elegant, because it turns a potentially complicated latent-and-selection problem into something that can still be linked back to standard discovery outputs. In that sense, the paper is not proposing “yet another search algorithm,” but rather a new semantics for reading the outputs of familiar methods, which I think is a real strength.

The main interpretation result is fairly crisp. In particular, the paper’s message that an oriented edge in the recovered CPDAG can be treated as a direct causal relation whose effect variable is not involved in selection, while an unoriented adjacency may still be explained by joint selection involvement, is a concrete takeaway rather than a vague theoretical statement. This gives the reader something operational to remember.

Weaknesses

The main weakness is that the experiments are a bit limited compared with how much the theory tries to cover. In the synthetic part, the results do show that the authors’ interpretation is better than the standard way of reading the graph, but the setup is also quite close to the model assumed in the paper. So for me, the synthetic results support the idea, but they do not fully answer how well the method would work when the real data do not match the assumptions so closely.

The reviewer also believes that some assumptions in the model are fairly simplified. In particular, the inheritance part seems to be modeled in a clean coordinate-wise way, which makes the theory easier to develop, but real biological or social processes are usually more complicated than that. The paper does mention this, but I still found myself wondering how much of the conclusion depends on this simplification.

For the real-data section, the figures are interesting and the interpretation is easy to follow, but the evidence is mostly based on whether the results look reasonable. I think this part would be stronger if there were one or two more concrete checks to support the discussion. Right now, I can see the intended message, but I am not fully convinced by the real-data results alone.

Another smaller point is that some of the theoretical discussion becomes dense in the middle of the paper. I could follow the main idea, but I think one more simple example would help, especially for the part explaining why the usual static-selection view is not enough and how the clique-augmented graph should be interpreted.

---

> ### Author Rebuttal · Authors · 2026-03-31
>
> We sincerely thank the reviewer for the insightful comments. Please see our response below.
>
> ---
> **(Q1)** For real-data experiments, the reviewer has concerns about i) some conclusions being phrased too strongly, and ii) the lack of quantitative evaluation.
>
> **A:** Thank you for these helpful points.
>
> For result interpretations, we have carefully adjusted the wording in the revision to clarify that they are only meant within the scope of our model assumptions.
>
> For quantitative evaluation, we actually share the same concern. We put a lot of efforts trying to find evolutionary-selection-related datasets with ground-truth causal graphs, but were unable to find any. That said, to make the evaluation more convincing, we have added the following analyses, with detailed results at **[[this anonymous link]](https://tinyurl.com/msyfxa8v)**.
>
> **1) A new dataset with partial ground truth.** We analyze an RNA-seq dataset on 422 genes from _Drosophila_. As partial ground truth, we use the 408 eQTL-probed (which may be incomplete) regulator–target pairs reported in [[E+20]](https://tinyurl.com/rbzys6dz). Using BOSS [[A+23]](https://tinyurl.com/bdd9ejct), the output CPDAG contains 469 oriented and 138 unoriented edges. We find that:
> - Oriented edges have a higher precision on causal adjacencies than unoriented edges (49% vs. 36%), consistent with our suggestion that under evolutionary selection, oriented edges are "more reliable" than unoriented ones in the recovered CPDAG.
> - Moreover, a gene appearing in many unoriented cliques is indeed reported to be related to fitness (e.g., insecticide resistance), which may also lend some support to our alternative interpretation of unoriented edges as joint selection involvement.
>
> **2) Evaluation with LLM-generated pseudo ground truth.** For the six datasets in our original submission, we provide variable names to ChatGPT and ask it to search the literature to annotate i) direct causal relations, and ii) variables directly involved in fitness selection. Using these annotations as _pseudo ground truth_, we evaluate:
> - Whether oriented edges in results are truly causal: We find that on 5 out of 6 datasets, oriented edges indeed have higher precision on causal adjacencies than unoriented edges.
> - Whether variables involved in selection (i.e., ancestors of $S$) truly form an unoriented clique: However, this is not as expected; the recall of such unoriented edges is consistently low (<= 50%), and we do not yet have a clear explanation.
>
> Of course, we are aware that the reliability of this procedure is not guaranteed, and we treat it only as a surrogate.
>
> ---
> **(Q2)** For synthetic experiments, the reviewer i) notes that the setup is close to the assumed model, and ii) suggests comparing with a latent-variable-oriented method.
>
> **A:** In light of your suggestions, we have expanded synthetic experiments with:
> 1. Comparisons to a latent-variable causal discovery method, RLCD [[D+24]](https://tinyurl.com/46may7xs); and
> 2. Results under model misspecifications with i) latent confounding, ii) dependent inheritance, and iii) structural changes across domains (edge reversals).
>
> We find that:
> - Though designed to model latent variables, RLCD still fails to recover the true causal relations under evolutionary selection.
> - While misspecifications degrade performance overall, our interpretation still outperforms the standard one, except in extreme cases where result graphs become overly dense.
>
> Detailed results are also available at **[[this anonymous link]](https://tinyurl.com/msyfxa8v)**.
>
> ---
> **(Q3)** The reviewer wonders how our model handles violations of the simplified _independent inheritance_ assumption.
>
> **A:** Thanks for this important point. Below we clarify which parts of our results extend, which do not, and what this implies for possible generalizations.
>
> When heritable factors $\epsilon$ are dependent (e.g., epistasis) and their effects on traits $X$ are non-component-wise (e.g., pleiotropy), our graphical model remains valid. In particular, Theorem 1 and Lemma 1 still hold: The evolutionary graph (with additional $\epsilon$-edges) still represents the CI relations in $X$, and evolution still induces only additional conditional dependencies, not independencies.
>
> However, these additional dependencies can no longer be represented by a clique-augmented DAG, so Theorem 2 and subsequent identification results no longer apply. While they can still be represented via MAG/FCI, as suggested by our experiments, such representations may be less informative in practice, due to fewer usable CI relations.
>
> Therefore, to improve identifiability, stronger and possibly equilibrium-aware parametric assumptions may be needed to exploit information beyond CI, as those already used in latent-variable causal discovery and latent state-space models. We hope that our model can serve as a useful starting point in such potential explorations.
>
> ---
> We thank the reviewer again for the insightful feedback.

---

> > ### Author Rebuttal · Reviewer_AMZw · 2026-04-03
> >
> > I'd like to keep the score unchanged.

---

### Official Review · Reviewer_JRS7 · 2026-03-18

**Soundness:** 2
**Presentation:** 3
**Significance:** 3
**Originality:** 3
**Overall Recommendation:** 5
**Confidence:** 3

**Summary:**

This paper proposes a formal graphical description for evolution induced selection (in contrast with the standard static selection formalism), shows that induced structure among observables are encoded in graphical notions already in use in causal discovery literature, and connects algorithms that estimate such structures with a novel interpretation for the proposed formalism.

**Compliance With Llm Reviewing Policy:**

Affirmed.

**Final Justification:**

My main (soundness) concerns are sufficiently cleared, and I have increased my rating accordingly from 3 to 5.

**Key Questions For Authors:**

Please see soundness section above for my list of "important" questions. If my concerns on those points are resolved, I am willing to improve my score.

Please add some direct comparisons and/or observations regarding the real data results.

Please extend limitations discussion.

Finally, why not model selection as a part of the generating causal system? The proposed formulation already is dynamical, and while selection is a conditioning in the static case, it might make more sense to model it as a dynamic driver/mechanism in your setting. I guess applying a final "select all currently alive" filter in *any* setting would be essentially equivalent to your current model but it would be a nice comparison to include.

**Limitations:**

Limitations discussion is limited.

- The model itself is too focused for the evolution use case: the only linkage between time slices is successful offspring generation (conditioning on S = 1) and a set of independently self-mutating exogenous variables. I do think it needs to be mentioned in the conclusion as a reminder.
- While the authors do some arguing on how faithfulness is justified, even if standard, it is not a light assumption.

**Strengths And Weaknesses:**

## Soundness

The claims made in the paper look, absent a couple parts I didn't fully follow, sound. However I do expect these parts to be clarified. Limitations are limitedly discussed. Assumptions used are standard in literature, and the authors demonstrate that their results still apply under slightly stronger structured assumptions (e.g. under invariant selection and inter time slice dependencies).

Finally, comparisons and/or qualitative analysis of real data experiments are weak.

Important notes:
- Proof of Lemma 1 is slightly weak. How historic selection d-separations imply d-separations in the static selection case is hard to follow.
- Theorem 1 altogether seems too extra. If the proof is a one liner, you don't nees to state (and analyze) the result in almost 1/2 column in the main paper.
- While I understand why the authors first provide Theorem A.1 then say proof of Theorem 2 is essentially the same thing (first establish/reference existing results), I believe moving said arguments to where "proof" of Theorem 2 is currently situated would make the proofs section easier to follow.
- I don't see how Theorem 4 proof follows from prior discussion (invariance of conditional distributions are introduced here for the first time, you must make the case why this follows from prior results, even a single reference suffices).
- Theorem 5 *statement* is slightly awkward.

A minor note:
- Some arguments don't actually generalize to multi-node S if it includes multiple independent components (e.g. in line 668, i and j don't have to be connected if S is disconnected itself).


## Presentation

Paper is cleanly written and clear. Literature comparisons look adequate. When mentioning parents of S are not identifiable, it is advisable to soften the language -- it is still possible that ancestors of S could be identified, and this would be the actual best case result in this paper's setiing and thus would be a *weaker* but still informative analysis.

## Significance

The authors deminstrate that their formalism is a non-equivalent way to formalize selection. The use cases used to motivate the approach indeed fits more naturally to their proposed framework. However, the model itself is too focused for the evolution use case: the only linkage between time slices is successful offspring generation (conditioning on S = 1) and a set of independently self-mutating exogenous variables. Given this is literally in the title of the paper I cannot be too negative about this limitation, but I do think it needs to be mentioned in the conclusion as a reminder.


## Originality

The formulation is novel, and methodology is very well connected to existing analyses in causal discovery literature. The "novel interpretation" part is well done and makes the contribution original enough.

---

> ### Author Rebuttal · Authors · 2026-03-31
>
> We sincerely appreciate the reviewer's constructive comments and helpful feedback. Please see below for our response.
>
> ---
> **(Q1)** The reviewer asks for clarification on several proofs:
>
> **(Q1.1)** _Lemma 1:_ How do evolutionary-selection d-separations imply static-selection d-separations?
>
> **A:** Thanks for this point. We have made the proof more explicit in the revision. Specifically, it is proved by contrapositive: For any d-connection in the _static graph_ $\mathcal{G}$, we can construct a corresponding open path in the _evolutionary graph_ $\mathcal{G}^{(T)}$, possibly by rerouting through previous time slices, and thus this d-connection still holds in $\mathcal{G}^{(T)}$.
>
> ---
> **(Q1.2)** _Theorem 1:_ It seems too extra.
>
> **A:** While the proof is simple, we include Theorem 1 mainly as a reference point in the overall logic. As a bridge from CI relations in the data to d-separations in the graph, it allows us to i) justify faithfulness (Corollary 1, which the 1/2 column discussion is actually about), and ii) further move to a simpler DAG representation (Theorem 2).
>
> We understand your concern, though, and have downgraded it from a "Theorem" to a "Lemma" in the revision.
>
> ---
> **(Q1.3)** _Theorem 4:_ How does the proof follow from prior results?
>
> **A:** We have made the proof more explicit in the revision: Instead of directly defining the _multi-domain clique-augmented DAG_, we first introduce an exogenous domain index variable (the "change driver") $\zeta$ into the evolutionary graph. Then, the invariance of a conditional distribution $p(X_A|X_C)$ can be expressed as the CI relation $\zeta \perp X_A|X_C$, which can further be read off from the corresponding d-separation in the evolutionary graph. Finally, we characterize these d-separations via a DAG, leading to Theorem 4.
>
> ---
> **(Q1.4)** Some arguments (e.g., line 668) do not generalize to multi-node $S$.
>
> **A:** Thank you for the careful reading. You are right. Our original phrasing ("..still hold with multiple selection variables.." in line 187) was intended to convey that the proof strategy extends, but the characterization can indeed differ. Specifically, with multi-node $S$, the induced CI relations can still be represented by a DAG with edges augmented among $S$'s ancestors; however, these edges now do not necessarily form a clique.
>
> We have clarified this point in the revision and added the separate result in this case in the appendix.
>
> ---
> **(Q1.5)** Other presentation issues.
>
> **A:** Thank you for these suggestions. We have clarified the _"statement"_ in Theorem 5 (previously omitted due to space limits), and reorganized the presentation by moving Theorem A.1 into the proof of Theorem 2.
>
> ---
> **(Q2)** The reviewer suggests adding more quantitative evaluation on real-world data.
>
> **A:** Thank you for this great point. We actually share the same concern, and put a lot of efforts trying to find evolutionary-selection-related datasets with ground-truth causal graphs for evaluation. Unfortunately, we were unable to find any.
>
> That said, to make the evaluation more convincing, we have added the following analyses:
> - Results on a new dataset with partial ground truth (DGRP RNA-seq);
> - Evaluation with LLM-generated pseudo ground truth on the original six real datasets; and
> - Extended synthetic experiments with additional baselines and under model misspecifications (e.g., dependent inheritance and latent confounding).
>
> Detailed results are available at **[[this anonymous link]](https://tinyurl.com/msyfxa8v)**. For more discussion, please kindly refer to our response to Reviewer yMH2's Q1.
>
> ---
> **(Q3)** The reviewer suggests extending the limitations discussion.
>
> **A:** Thank you for this suggestion. In the revision, we have carefully extended the limitations discussion on the following aspects:
> - The model's specificity to the evolution use case;
> - Causal sufficiency;
> - Faithfulness (yes, we totally agree that it is not a light assumption);
> - Independent inheritance (see our response to Reviewer yMH2’s Q2); and
> - Cross-domain structural invariance (see our response to Reviewer 4yqK's Q1).
>
> ---
> **(Q4)** The reviewer wonders whether selection can be modeled as part of the generating causal system.
>
> **A:** Thank you for the question. We are not sure if we fully understand your point; if the following response differs from what you intended to mean, please kindly let us know.
>
> In our current modeling, the selection variable $S$ can also be viewed as a "dynamic driver", i.e., $S^{(t)}$ pointing to $X^{(t+1)}$, with semantics like "progenitor fitness affects offspring mechanisms". However, since $S^{(t)}$ is then conditioned on a constant value (we only observe individuals whose progenitors reproduce successully), the effects of these $S^{(t)}\to X^{(t+1)}$ edges vanish in the data. As a result, the model reduces back to the standard conditioning way.
>
> ---
> Thank you again for all the valuable feedback, and we hope your questions are properly addressed.

---

> > ### Author Rebuttal · Reviewer_JRS7 · 2026-04-01
> >
> > My main (soundness) concerns are sufficiently cleared, and I have increased my rating accordingly from 3 to 5.

---

### Official Review · Reviewer_yMH2 · 2026-03-22

**Soundness:** 3
**Presentation:** 4
**Significance:** 3
**Originality:** 4
**Overall Recommendation:** 5
**Confidence:** 4

**Summary:**

Standard graphical models for selection assume static, one-shot filtering. In evolutionary systems (e.g., antibiotic resistance, immune adaptation, social norm emergence), observed data does not represent the global population because its properties evolve through repeated rounds of differential fitness and reproduction. Static selection models cannot capture this.

The authors formalize evolutionary selection as a multi-generational DAG (Definition 1), where each generation's traits depend on inherited exogenous factors, within-generation causal mechanisms, and selection-driven reproduction. The central technical result is the clique-augmented DAG G+ (Definition 2, Theorem 2): the CI constraints under arbitrary-generation evolutionary selection reduce to d-separations in a simple DAG over the observed variables alone, regardless of the generation T. This is what makes the practical algorithm possible -- standard PC/GES apply directly (Algorithm 1), with a reinterpreted edge semantics: directed edges represent true causal relationships, while undirected edges may reflect joint involvement in selection. A key implication: evolutionary selection can only be falsified, never confirmed from data -- fundamentally different from the static case.

The theoretical chain is clean: Theorem 1 (Markov properties), Corollary 1 (no extra CIs even under mechanism invariance), Theorem 2 (G+ representation), Theorem 3 (soundness and completeness of Algorithm 1), and Theorems 4-5 (multi-domain extension via CDNOD). Three scenarios in Section 3.2 show what goes wrong under incorrect approaches: (1) ignoring selection entirely, (2) modeling it as static via FCI, (3) using MAG-FCI even with the correct evolutionary model.

The authors validate on synthetic data (random Erdos-Renyi graphs, d in {10, 15, 20}, average degree 2, d/5 selection parents, linear SEM) and 6 real-world datasets: Cranial (human cranial measurements), Panzea (maize morphology), PanTHERIA (mammal traits), AVONET (bird phenotypes), CSES (political survey), and PUMS (U.S. Census).

**Compliance With Llm Reviewing Policy:**

Affirmed.

**Key Questions For Authors:**

1. Can you run standard PC and evolutionary-PC on the same real-world datasets and show where they differ? The infrastructure already exists for synthetic data (Figure 6). Even without ground truth, a domain-informed comparison of the outputs would be convincing.

2. How would the framework handle heritable factors that do not operate independently? Is there a natural extension, or does the independence assumption reflect a fundamental constraint of the model?

**Limitations:**

The paper acknowledges the independence assumption for heritable factors (Section 6), noting pleiotropy and epistasis as violations. Beyond this, two issues go unaddressed: quantitative benchmarks run only on synthetic data, and no side-by-side comparison of standard PC vs. evolutionary-PC on real-world datasets exists. The infrastructure for such a comparison already exists (Figure 6 does it for synthetic data), so extending it to real data should be straightforward.

**Strengths And Weaknesses:**

## Strengths

- **Novel formalization.** The authors uniquely formalize why static models in causal discovery fail when properties evolve over time, and propose a new modeling framework to address this.
- **Elegant central result (Theorem 2).** The clique-augmented DAG G+ proves that arbitrarily complex multi-generational evolutionary graphs reduce to a simple DAG over observed variables. This is technically clean and practically powerful -- it makes Algorithm 1 possible without any new inference machinery.
- **Five theorems building in a clean logical chain.** Theorem 1 through Theorem 5, plus Corollary 1, form a tight progression from Markov properties to multi-domain identification. Each result builds on the previous one.
- **Excellent pedagogical scenarios (Section 3.2).** Three scenarios show exactly what goes wrong under three incorrect approaches (ignoring selection, modeling it as static, using MAG-FCI). These make the paper's contribution concrete and help the reader understand why existing tools fail.
- **Sound definitions, theorems, and proofs.** The theoretical foundations are rigorous. Examples help readers unfamiliar with causal graph theory understand why static modeling falls short.
- **Elegant reuse of existing algorithms.** Standard PC/GES algorithms apply directly with reinterpreted edges. This makes the framework practical without requiring new inference machinery.
- **FAQ sections preempt reader objections.** Useful for a theory-heavy paper.
- **Well-designed, informative figures.**
- **Broad applicability.** Evolutionary properties appear across many fields. The 6 diverse real-world datasets (spanning archaeology, agriculture, ecology, ornithology, political science, and demography) illustrate this breadth.

## Weaknesses

1. **Quantitative benchmarks only on synthetic data.** Figure 6 shows precision metrics for d=20 observed variables, comparing the standard vs. evolutionary interpretation of PC/GES output. Real-world datasets lack ground-truth graphs, but this leaves the reader unable to judge the model's superiority on real problems.
2. **No side-by-side comparison of standard PC vs. evolutionary-PC on real-world data.** The comparison already exists for synthetic data (Figure 6) -- the infrastructure is in place, so extending it to real data should be straightforward. Running both algorithms on the same real-world datasets and showing where they differ would be convincing -- e.g., "standard PC claimed X_1 causes X_2, but they are jointly involved in coselection." Without this, the "aligns with domain knowledge" arguments in Figure 5 seem one-sided. This is the single change that would most strengthen the paper.
3. **Independence assumption for heritable factors constrains applicability.** The authors acknowledge this (Section 6), noting pleiotropy and epistasis as violations. In complex biological systems (e.g., bacterial communities), multiple interacting factors often govern traits. The justification is reasonable, but this does limit the model's reach. A brief discussion of how the framework might extend to dependent heritable factors -- even speculatively -- would help readers gauge the model's long-term potential.
4. **Figure 5 shows only example subgraphs.** The reader must go to the appendix to assess real-world applicability. Including the full graphs in the main text (or at least graph-level statistics) would help.
5. **Complex genomic relationships untested.** The limited quantitative evaluation leaves open questions about how the model handles high-dimensional, multi-factorial biological data. For a paper with strong theoretical contributions, this is acceptable, but it would strengthen significance.

---

> ### Author Rebuttal · Authors · 2026-03-31
>
> We sincerely thank the reviewer for the insightful comments. Please see below for our response.
>
> ---
> **(Q1)** The reviewer has concerns about the lack of quantitative evaluation on real-world data.
>
> **A:** Thank you for this great point. We actually share the same concern, and put a lot of efforts trying to find evolutionary-selection-related datasets with ground-truth causal graphs for evaluation. Unfortunately, we were unable to find any.
>
> That said, to make the evaluation more convincing, we have added the following analyses, with detailed results available at **[[this anonymous link]](https://tinyurl.com/msyfxa8v)**.
>
> **1) A new dataset with partial ground truth (DGRP RNA-seq).** We analyze an RNA-seq dataset on 422 genes from the _Drosophila melanogaster_ Genetic Reference Panel (DGRP). As partial ground truth, we use the 408 eQTL-probed (which may be incomplete) regulator–target pairs reported in [[E+20]](https://tinyurl.com/rbzys6dz). Using BOSS (a scalable causal discovery method; [[A+23]](https://tinyurl.com/bdd9ejct)), the output CPDAG contains 469 oriented and 138 unoriented edges. We find that:
> - Oriented edges have a higher precision on causal adjacencies than unoriented edges (49% vs. 36%), consistent with our suggestion that under evolutionary selection, oriented edges are "more reliable" than unoriented ones in the recovered CPDAG.
> - Moreover, a gene appearing in many unoriented cliques is indeed reported to be related to fitness (e.g., insecticide resistance), which may also lend some support to our alternative interpretation of unoriented edges as joint selection involvement.
>
> We hope that this experiment can also partly address your concern on _"complex genomic relationships untested"_.
>
> **2) Evaluation with LLM-generated pseudo ground truth.** For the six real-world datasets in our original submission, we provide the variable names to ChatGPT and ask it to search the literature to annotate i) direct causal relations, and ii) variables directly involved in fitness selection. Using these annotations as _pseudo ground truth_, we evaluate our results in the following two aspects:
> - Whether oriented edges are truly causal: We find that on 5 out of 6 datasets, oriented edges indeed have higher precision on causal adjacencies than unoriented edges.
> - Whether variables involved in selection (i.e., ancestors of S) truly form an unoriented clique: However, this is not as expected; the recall of such unoriented edges is consistently low (<= 50%), and we do not yet have a clear explanation.
>
> Of course, we are aware that the reliability of this procedure is not guaranteed, and we treat it only as a surrogate.
>
> **3) Extended synthetic experiments.** We further enhance the synthetic study with i) comparisons to a latent-variable causal discovery method, and results under model misspecifications with ii) latent confounding, iii) dependent inheritance, and iv) structural changes across domains (edge reversals). We find that:
> - The used latent-variable method also fails to recover the true causal relations under evolutionary selection.
> - While model misspecifications degrade performance overall, our interpretation still outperforms the standard one, except in extreme cases where result graphs become overly dense.
>
> Taken together, while fully reliable ground truth remains unavailable for real data, we hope that these additional analyses can help partly address your concern.
>
> ---
> **(Q2)** The reviewer wonders how our framework would handle violations of the simplified _independent inheritance_ assumption.
>
> **A:** Thanks for this important point. Below we clarify which parts of our results extend, which do not, and what this implies for possible generalizations.
>
> When heritable factors $\epsilon$ are dependent (e.g., epistasis) and their effects on traits $X$ are non-component-wise (e.g., pleiotropy), our basic graphical modeling remains valid. In particular, Theorem 1 and Lemma 1 still hold: The evolutionary graph (with additional $\epsilon$-edges) still represents the CI relations in $X$, and evolutionary selection still induces only additional conditional dependencies, but not independencies.
>
> However, these additional dependencies can no longer be represented by a simple clique-augmented DAG, and thus Theorem 2 and subsequent identification results no longer apply. While they can still be represented by the MAG/FCI framework, as suggested by our additional experiments, such representations may be less informative in practice, as fewer CI relations remain usable.
>
> Therefore, to improve identifiability, stronger and possibly equilibrium-aware parametric assumptions may be needed to exploit information beyond CI, as those already used in latent-variable causal discovery, causal representation learning, and latent state-space models. We hope that our graphical model can serve as a useful starting point in such potential explorations.
>
> ---
> We want to thank the reviewer again for the insightful feedback.

---

> > ### Author Rebuttal · Reviewer_yMH2 · 2026-04-03
> >
> > Thank you for the strong rebuttal. I recommend this paper for acceptance and maintain my current score (accept).

---

### Decision · Program_Chairs · 2026-04-30

**Decision:**

Accept (spotlight)

**Comment:**

The paper has received consistently strong evaluations, and the rebuttal further strengthened confidence in the work by addressing the main reviewers’ concerns. In particular, reviewers found the central distinction between static and evolutionary selection to be novel and important, the theoretical development to be sound and elegant, and the resulting interpretation of standard causal discovery outputs to be both clear and practically meaningful within the paper’s setting. While some limitations remain regarding empirical breadth and modeling scope, these were acknowledged and do not outweigh the paper’s substantive conceptual and technical contributions. Overall, I believe this paper can be a great addition to ICML’s program and I recommend it for publication.